# A Decade of Variability on Jakobshavn Isbrae: Ocean Temperatures Pace Speed Through Influence on Mélange Rigidity

Ian Joughin[1], David E. Shean[2], Benjamin E. Smith[1], Dana Floricioiu[3]

[1]Applied Physics Laboratory, University of Washington, Seattle, 98105, USA
[2]Department of Civil and Environmental Engineering, University of Washington, Seattle, 98185, USA
[3]German Aerospace Center (DLR), Remote Sensing Technology Institute, SAR Signal Processing, Muenchenerstr. 20, 82230 Wessling, Germany

[1]Department example, University example, city, postal code, country
[2]Laboratory example, city, postal code, country

*Correspondence to*: Ian Joughin (ian@apl.washington.edu)

**Abstract.** The speed of Greenland's fastest glacier, Jakobshavn Isbrae, has varied substantially since its speedup in the late 1990s. Here we present observations of surface velocity, mélange rigidity, and surface elevation to examine its behaviour over the last decade. Consistent with earlier results, we find a pronounced cycle of summer speedup and thinning followed by winter slowdown and thickening. There were extended periods of rigid mélange in the winters of 2016–17 and 2017–18, concurrent with terminus advances ~6 km farther than in the several winters prior. These terminus advances to shallower depths caused slowdowns, leading to substantial thickening, as has been noted elsewhere. The extended periods of rigid mélange coincide well with a period of cooler waters in Disko Bay. Thus, along with the relative timing of the seasonal slowdown, our results suggest that the ocean's dominant influence on Jakobshavn Isbrae is through its effect on winter mélange rigidity, rather than summer submarine melting. The elevation time series also reveals that in summers when the area upstream of the terminus approaches flotation, large surface depressions can form, which eventually become the detachment points for major calving events. It appears that as elevations approach flotation, basal crevasses can form, which initiates a necking process that forms the depressions. The elevation data also show that steep cliffs often evolve into short floating extensions, rather than collapsing catastrophically due to brittle failure. Finally, summer 2019 speeds were slightly faster than the prior two summers, leaving it unclear whether the slowdown is ending.

## 1 Background

Except for a few brief intervals, Jakobshavn Isbrae (see location in Fig. 1) has been the fastest and largest (by discharge volume) glacier in Greenland over the last several decades (Mouginot et al., 2019). In the mid 1990s, it thickened slightly after slowing down relative to the 1980s (Joughin et al., 2004; Thomas et al., 2003). By the late 1990s and early 2000s, however, it

began to speed up as its floating ice tongue disintegrated (Joughin et al., 2004; Krabill et al., 2004; Luckman and Murray, 2005). Although the glacier maintained a steady speed year round in the 1980s (Echelmeyer and Harrison, 1990), following its speedup, a strong seasonal variation in speed developed in response to the annual advance and retreat of its calving terminus (Joughin et al., 2008a; 2008b). The increased extension during the summer speedup produced near-terminus dynamic thinning of ~30 m/yr, which was partially offset by ~15 m/yr of thickening as the glacier slowed over the winter (Joughin et al., 2012b). While the speed peaked in the summer of 2012, similar large speedups continued over the next several summers (Joughin et al., 2018; 2014). Over the winter of 2016-2017, however, the terminus began to slow (Lemos et al., 2018), leading to a peak speed during the summer of 2017 that was comparable to the minimum speeds for several of the previous winters (Joughin et al., 2018).

With this slowdown, the terminus region transitioned from net annual thinning to net annual thickening (Khazendar et al., 2019). This slowdown coincided with a period of increased surface mass balance and cooler atmospheric and ocean temperatures (Joughin et al., 2018; Khazendar et al., 2019; Mouginot et al., 2019). While it is clear that seasonal variation modulates Jakobshavn Isbrae's flow speed, the causal links and sensitivity to underlying atmospheric and oceanic forcing remain poorly understood. Here we investigate the nature of the recent changes and investigate potential links to recent climate variability.

While surface velocity is reasonably well sampled over the last decade, past studies have been limited by sparse spatial and temporal sampling of the large and rapid changes in Jakobshavn Isbrae's surface geometry. To help fill this gap, we assembled a comprehensive time series of digital elevation models (DEMs) from several optical and radar sensors. Together, the velocity and elevation data sets provide a detailed look at the evolution of Jakobshavn Isbrae's terminus region over the last decade.

## 2 Methods

We used data from a variety of radar and optical sensors to determine terminus position, surface velocity, elevation, and other conditions in the fjord over the last decade.

### 2.1 Terminus Position Data

Using several hundred TerraSAR-X images with ~20-m resolution, we manually digitized the points where the profile shown in Fig. 1 crosses the terminus, as a proxy for the overall terminus position. Because of the rapidly changing elevations relative to the static DEM used for geolocation, the absolute geolocation errors of points at the terminus could be as large as ~100 m. Since we used images acquired from both ascending and descending orbits, the relative error could be nearly twice as large between images, because a given height error will induce geolocation error in opposite directions (i.e., the terminus could appear up to nearly 200 m apart in images acquired at the same time from ascending and descending orbits, even though

individually each would be in error by only 100 m). Thus, these data are used to examine the overall seasonal pattern of advance and retreat, rather than to identify the calving of individual icebergs, which are typically about 100–200 m in the along-flow dimension (Amundson et al., 2010).

## 2.2 SAR Derived Velocity and the Presence of Rigid Mélange

We determined surface velocity over the fast moving region of Jakobshavn Isbrae for the period from January 2009 through mid-November 2019 by applying standard speckle-tracking methods (Joughin, 2002) to stripmap TerraSAR-X and TanDEM-X synthetic aperture radar (SAR) data provided by the German Aerospace Center (DLR). We also applied these algorithms to

70 several Cosmo-SkyMed images from the summer of 2019, which were provided by the Italian Space Agency (ASI).

The formal surface velocity errors are small ($\sim< 10$ m/yr) relative to the speeds we examine here (>1000 m/yr), so the precision is good with respect to points collected using the same imaging geometry and general time period. Because we make the measurements assuming the flow is parallel to the surface (Joughin et al., 1998), however, errors in the surface slope can

introduce absolute errors of ~0.5–3%. With the relatively accurate DEMs we used, the errors should tend toward the low end of this range in most instances. Exceptions may occur where large temporal fluctuations in slope occur near the terminus as discussed below. As with the SAR image data, horizontal geolocation errors are an issue, although to a lesser extent because the DEM is updated annually for velocity. Despite the annual DEM updates, large intra-annual elevation changes can introduce absolute geolocation errors to the velocity products of up to ~50 m. In such cases, an otherwise correct velocity measurement

is posted at the wrong location, which in a gridded product is equivalent to an additional source of velocity error, especially where velocity gradients are strong. This problem can be exacerbated when comparing data acquired from differing imaging geometries (e.g., from ascending and descending passes), since the DEM-induced location shifts can occur in opposing directions to produce a relative geolocation error of ~100 m.

In addition to the glacial ice, the speckle tracker was also applied to the regions in the fjord where an ice mélange (mixture of sea ice and icebergs) is often present. When the mélange is rigid and moves within the bounds of the tracking algorithm (i.e., at speeds similar to the terminus), a successful velocity estimate is returned. In such cases, the velocity is indicative of rigid mélange that is being pushed down the fjord by the advancing terminus (Joughin et al., 2008b). Absence of an estimate usually indicates non-rigid motion in which there is substantial relative motion within the ~200-m dimensions of the tracking window,

yielding little correlation between image patches acquired several days apart. Alternatively, the trackers failure to produce a result could indicate strong rotation or extremely fast rigid translation down the fjord (e.g., large calving event or wind-driven advection) at rates beyond what tracker was designed to accommodate (i.e., much faster than the terminus). For the remainder of the text, when we refer to 'rigid mélange,' we mean a mélange that is being coherently advected at a rate similar to the flow speed at the terminus.

## 2.3 Terminus Elevation Data

We used elevation data from a number of sources including both optical and radar sensors. All elevations are given as height in meters above the EGM2008 geoid, which approximates mean sea level. All DEMs were up- or down-sampled to a consistent 25-m posting for analysis.

We generated DEMs from DigitalGlobe WorldView-1/2/3 (WV) and GeoEye-1 (GE) imagery acquired between June 2008 and May 2015 using the NASA Ames Stereo Pipeline (Beyer et al., 2018) and methods outlined in Shean et al. (2016). The WV/GE DEMs are posted at 2 m, with a swath that is ~13-17 km wide and ~13-110 km long, depending on sensor and acquisition mode. The DEMs were co-registered to available ground control points (GCPs from NASA's ICESat, LVIS and ATM lidars) over static control surfaces (e.g., exposed bedrock) using a rigid-body iterative closest-point algorithm to determine the translation that minimized the elevation difference between the DEMs and GCPs (Shean et al., 2016). After co-registration, the residual error of each DEM was estimated using the normalized median absolute deviation (NMAD; Hoehle and Hoehle, 2009) of all GCP-DEM differences (Shean et al., 2016), with values for WV DEMs of 0.52 m.

We used ArcticDEM strips also derived from WV/GE imagery acquired between 2016–2019 (Porter et al., 2018), generated using the SETSM processing workflow (Noh and Howat, 2017). The elevation of each ArcticDEM was adjusted by adding a constant offset to produce a self-consistent set of results that minimized biases in regions of overlap on bedrock. After this procedure, comparison with several thousand independent lidar point measurements over exposed bedrock yielded a median bias of 1.2 m with median standard deviation of 0.9 m.

We included DEMs from TanDEM-X SAR images acquired between 2011–2013 in bistatic, stripmap mode and processed using the Integrated TanDEM-X Processor (ITP) (Rossi et al., 2012). The DEM posting was 5 m with a ~35 km wide and ~50-60 km long footprint. Absolute elevation offsets from baseline-dependent interferometric SAR (InSAR) height ambiguities were corrected by adjusting the absolute phase offset during InSAR processing. We further improved the accuracy by applying horizontal and vertical adjustments constrained by ground control points with a similar procedure to that applied to the WV/GE data described above (Shean et al., 2016), yielding accuracy of 1.2 m.

We obtained four DEMs produced by the Glacier and Land Ice Surface Topography Interferometer (GLISTIN) collected in each spring from 2016 to 2019, which are publicly available from the JPL Uninhabited Aerial Vehicle SAR (UAVSAR) website (http://uavsar.jpl.nasa.gov). The GLISTIN system uses Ka-Band to limit penetration in ice and has meter-scale accuracy (Moller et al., 2011), which is comparable to the other DEMs we used in this study.

**2.4 Flotation Height**

We computed the flotation height, $h_f$, equal to the maximum height that the surface of a column of floating ice could attain without its bottom touching the bed, using standard methods and assuming densities of 910 and 1028 kg/m³ for ice and sea water, respectively (Cuffey and Paterson, 2010). The bed data used in this calculation are from BedMachine Version 3 (Morlighem et al., 2017). Estimated bed elevation uncertainty for this region is up to ~100 m, which translates to an uncertainty of 13 m in $h_f$.

**3 Results**

As described throughout this section, we have assembled a comprehensive set of terminus position, velocity, and elevation data that provide a detailed view of Jakobshavn Isbrae's behaviour over the last decade.

**3.1 Terminus Position**

The top panel in Fig. 2 shows the terminus position (black points) along our profile (Fig. 1) as a function of time. Also shown are the points along the profile located 1-km behind the point of maximum annual terminus retreat for each year ($T_{max}$-1km; brown points). In determining $T_{max}$-1km for 2014, we excluded two anomalous points associated with rapid retreat and re-advance, which appear to have been from an area that was floating and had little impact on the speed. As in prior time series (Joughin et al., 2014), these data show seasonal variation in terminus extent by more than 2 km, with the terminus retreating farther inland each year through 2012. From 2012 through 2016, the extent of maximum retreat was similar each summer as indicated by the $T_{max}$-1km positions (brown lines Fig. 2). The seasonal transition from advance to retreat typically occurred around March, but it started as early as January (2010) and as late as June (2011). Peak retreat generally occurred in late summer/early fall.

In the winter of 2008–09, a several-kilometre long transient ice tongue advanced, as was the case for the prior winters following the breakup of the more extensive ice tongue (Joughin et al., 2012b). This pattern was altered for the 2009–10 through 2015–16 winters when there was no substantial ice-tongue advance, although, as described below, the terminus was floating at times. The pattern changed again when a floating tongue 4–5 km in length advanced during the winter of 2016–17 (see Fig. 2), with an even longer ice tongue advancing the following winter. An ice tongue also advanced in the 2018–19 winter, but it was less extensive than those from the prior two winters.

**3.2 Mélange Extent**

We sampled the mélange in a static, 1 km x 1 km region just outside the point of maximum terminus advance for 2010–2019. Because the extent of the floating ice tongue was so much greater in 2009, we moved the sampling region for the 2008–09

winter farther down the fjord (see blue boxes Fig. 1). The blue symbols in the top panel of Fig. 2 indicate the times when rigid mélange was detected in front of the terminus. This detection of rigid mélange is relatively robust, hence there should be few false detections (positives). Missed detections could result if the speckle-tracker fails due to poor correlation despite the presence of rigid mélange. We examined the individual maps by hand, and there are few, if any, instances of missed detections. As plotted, a gap in coverage would also indicate a lack of mélange. Such gaps are indicated by a corresponding lack of velocity data in the lower panel of Fig. 2 (e.g., gap in Fall 2010).

The duration of periods with rigid mélange varies substantially from year to year. The longest periods of rigid mélange occurred during the 2016–17 through 2018–19 winters, which coincide well with the periods when a more extended floating ice tongue developed. Similarly, there was extensive mélange in the 2008-2009 winter when a floating tongue also advanced (only 2009 is shown in Fig. 2). For other years the mélange generally was less frequent and more sporadic. The occurrence of rigid mélange was particularly infrequent in both 2011 and 2012, which are the years when the greatest retreat occurred. In all summers there was a rigid-mélange-free period of at least a few months, which tends to line up well with the periods of summer speedup. In general, extended periods where rigid mélange exists coincide well with periods of terminus advance, which is consistent with similar data from the period from 2005 to 2007 (Joughin et al., 2008b).

**3.3 Velocity**

Figure 2 (bottom panel) shows the speed at several points along the main trunk, which represents an update to earlier records (Joughin et al., 2014; 2018). For consistency, we use the same locations and naming schemes (M6–M26) as earlier work (Joughin et al., 2008b; 2014). We also include the speed at the point 1-km upstream of the point of maximum annual terminus retreat ($T_{max}$-1km; see brown Fig. 2 top), which provides speed relative to the evolving terminus position rather than a fixed point. As noted elsewhere (Joughin et al., 2014), when the terminus migrates inland along a reverse bed slope, the speed at a fixed point should increase due to the reduced proximity to the steep face of the thicker calving front (Joughin et al., 2014). Thus, the $T_{maxn}$-1km time series is much faster than the M6 data in the early part of the record when the terminus was more advanced, but the speeds for the two points begin to converge as the terminus migrates inland. From 2012–2016 the speeds at these two points are nearly identical, consistent with their similar proximity to the terminus, with separation by ~1-km in the cross-flow direction (see Fig. 1).

Figure 2 indicates that the slowdown that began in the winter of 2016–17 (Joughin et al., 2018; Lemos et al., 2018) continued, with the slowest summer speeds for the 2009–2019 period occurring in 2018. Although the 2018–19 winter speeds were similar to the previous winter, the terminus retreated rapidly beginning in late April 2019 with an accompanying increase in speed. Later the following summer, the peak speed was ~1500 m/yr faster than in summer 2018, but comparable to that in summer 2017.

## 3.4 Elevation

We assembled a total of 85 DEMs spanning the period from 2010 to 2019 that incorporates data from ASP- (50) and SETSM- (20) processed WorldView stereo pairs, TanDEM-X acquisitions (11), and GLISTIN (4) airborne campaigns. Figure 2 (middle) shows the full elevation time series at points M6 and M9. At these points, surface was highest in late spring 2011 after winter thickening recovered some of the loss that occurred the prior summer. The surface then lowered by ~30 m during the 2011 summer, but thickened moderately (~10 m) during the subsequent winter. As the terminus retreated in summer 2012, the surface at M6 dropped by nearly 40 m between June 10 and July 20, coincident with the most extreme speedup event in the entire record. From summer 2012 to summer 2015, there was a fairly consistent annual cycle with the terminus thickening by 35–50 m each winter, and thinning by a similar amount each summer, bringing M6 to near or at flotation in 2013–2015. The record is sparser for 2016, but it is sufficient to indicate that increased thickening occurred during the winter of 2016-17 as the ice tongue advanced and the glacier slowed. As a result, just prior to the summer speedup in 2017, the terminus region was ~35 meters higher than the previous summer (see also Khazendar et al., 2019). Winter thickening continued to outpace summer thinning so that by March 2019, surface elevations near the terminus were only 10 to 20 m lower than those observed during spring of 2011 (Fig. 2 middle).

Figure 3 shows the surface elevation time series along the profile shown in Fig. 1 along with the flotation height ($h_f$ ; solid grey) computed using BedMachine v3 data (Fig. S1; Morlighem et al., 2017). For reference, the first profile (August 12, 2010) is plotted (black circles) for comparison with each year. The plot for 2018 also includes three 2019 profiles (see also Fig. S2 for separate 2018 and 2019 plots). In general, the profiles show the same evolution described above for the M6 and M9 points. The spring 2019 profiles are similar to the August 2010 profile, indicating some degree of recovery in response to recent slowdown and thickening (Joughin et al., 2018; Khazendar et al., 2019). For the region within a few kilometres of the terminus, there was ~15–35 thinning in the summer of 2019, which was sufficient to offset the 2018–19 winter thickening to yield annual thinning of ~10–20 m from October 2018 to October 2019 (see 2018 plot Fig 3 and Fig. S2), indicating a resumption of annual thinning (at least for this period).

The data reveal a variety of terminus geometries. Several of the profiles in Fig. 3 show a relatively steep calving face in mid to late summer (e.g., August 17, 2010). Another common configuration occurs where the elevation ~0.5–2 km upstream of the terminus is well below $h_f$, indicating the presence of an ephemeral floating ice tongue. At times when the terminus was afloat, there were cases when the elevation tapered smoothly to the water surface with little or no terminal cliff (e.g., August 12, 2010). In many other cases, particularly in many of the early spring profiles, the downstream part of the floating region was relatively flat with a relatively steep calving front.

Between 2012 and 2015 in late summer there were often ~1–2 km transverse-width depressions with surface heights tens of
meters below flotation located a kilometre or more upstream of the well-grounded terminus. There are insufficient data to
determine whether such depressions formed in 2016, though depressions that remained above flotation are present in the March
through July 2016 profiles. Similar features are not present in the 2017–2019 data, which appear similar to the 2010 and 2011
summer data when the surface also was well above flotation. When they are present, these depressions advect downstream
with the ice flow, as is the case for similar features that have been observed elsewhere (James et al., 2014; Joughin et al.,
2008c).

To provide a better picture of the spatial pattern of terminus variation, Fig. 4 shows a subset of the DEMs. In addition to the
colour scale to represent elevation, the 100- (white) and 150-m (black) contours are also shown. Below ~100 m, most of the
ice in the main trunk should be fully afloat, and above 150 m it should be fully grounded. The region between these contours
represents the transition from grounded to floating ice, including the grounding line. As with the profile data, the map-view
data reveal periods when the terminus is an abrupt grounded cliff, often indicated by areas where the 100-m contour tracks the
terminus. At other times, a heavily crevassed but generally intact floating extension is clearly visible. In particular, the extended
tongues that formed during the 2016–17 and 2017–18 winters are still visible in the respective early spring DEMs.

The DEMs in Fig. 4 indicate complex spatial and temporal variations in the near terminus geometry. There is a persistent
depression that forms in the lee of what appears to be a bed constriction on the north side of the main channel (see area around
blue diamond plotted on the August 2010 DEM shown in Fig. 4). Relatively little thinning occurs on the steep slope
immediately upstream of this feature, leading to little variation in the position of 100 and 150-m contours.  Towards the centre
of the main trunk, however, the contours migrate back and forth ~2.5 km over the observation period, which is consistent with
the point data shown in Fig. 2. This variation can yield large changes in the near terminus slope. For example, in the July 2012
DEM, the two contours are separated by less than 500 m. By contrast, in the April 2014 DEM the 150 m contour intersects the
profile at nearly the same point (black diamond in Fig. 4), but the 100-m contour is located nearly 2-km farther downstream.
On the south side of the fjord, there is a ridge (e.g., December 2011) that separates a pair of local topographic lows. Over the
course of the record, this area evolves substantially as the surface elevation varies by several 10s of meters. There are times
when the upstream depression extends inland so that the south side of the channel appears to reach, or at least approach,
flotation while the centre and north side remain well grounded.  Toward the centre of the main trunk, the ephemeral surface
depressions mentioned above are visible in the September 2013 and August 2014 DEMs (see closed contours between the
black diamond and star in Fig. 4). For many other times, instead of a depression, this region represents a cross-flow high for
the main trunk.

## 4 Discussion

The results described above provide a detailed view of the evolution of the terminus region of Jakobshavn Isbrae over the last decade. Here we provide additional interpretation, including discussion of the relation between speed and bed topography; how ocean temperature may influence the seasonal speedup through its control on mélange rigidity; the processes that produce surface depressions and related calving events; and ice cliff stability.

### 4.1 Relation Between Speed and Terminus Position

As with previous work on Jakobshavn Isbrae (Joughin et al., 2012b), Fig. 2 reveals a strong correlation between seasonal speedup and terminus position. To investigate further, Fig. 5 shows year-by-year linear regressions of terminus speed ($T_{max}$-1 km) against terminus position. We limited the regressions to terminus positions >7500 m along the profile, which excludes many of the locations where the terminus was clearly floating and providing little resistance. For all but 2009 and 2010, a linear relation between the two explains >75% of the variance ($r^2$=0.76–0.92). The lowest correlation occurs in 2010 ($r^2$=0.32), which was a year when there was relatively little retreat and corresponding speedup, hence other factors may dominate.

If we consider the terminus position as a proxy for the grounded terminus depth during retreat/advance along a retrograde slope, then the results in Fig. 5 are consistent with increasing terminus thickness producing speedups during periods of retreat, and vice versa during advance. While this relation should be non-linear (Howat et al., 2005; Thomas, 2004), it appears to be relatively linear over the scale of the annual terminus-position perturbations (~3 km) as indicated by the $r^2$ values. Despite the good agreement, it is important to note that there is still significant speed variation not accounted for by terminus position. In particular, some of the calving events that rapidly affected terminus position likely occurred from a floating or lightly grounded terminus, which should yield little or no speedup (Cassotto et al., 2019). For example, the brief period of retreat and rapid re-advance in the autumn of 2014 with no corresponding speedup (Fig. 2) was likely due to calving from a floating ice tongue. In addition, variation in the width of the channel and local bed topography should also influence speed as the terminus retreats inland.

Both the slope and intercept of the regressions vary substantially over time (Fig. 5). The variation in slope to some extent may reflect the expected non-linearity noted above, with increasing sensitivity as the terminus retreats into deeper water. Faster speeds at less retreated positions earlier in the record (2009–2011) may be due to the generally thicker terminus region (see M6 and M9 elevations in Fig. 2), which should yield a greater surface slope and driving stress in the region upstream of the terminus.

## 4.2 Influence of Ocean and Atmosphere Temperatures on Glacier Speed and Geometry

Figure 6 shows ocean temperature (similar to that from Khazendar et al., 2019) and winter air temperature anomalies for the period from 1980 to the most recent observations. The strong summer speedups from 2012–2015 (Fig. 2) coincide with periods when the water in Disko Bay was relatively warm (>3ºC). Conversely, the slowdown that began in Fall 2016 took place when the water in Disko Bay was relatively cool (~1.5ºC). Based on this relation, it has been argued that enhanced melting of the terminus produced greater calving, retreat, and speedup, particularly in summer when buoyant subglacial meltwater plumes

should enhance circulation at the terminus (Khazendar et al., 2019). While water temperature in the fjord should undoubtedly modulate melt, it is less clear that this apparent correlation establishes a causal relation between enhanced melt at the terminus and the observed speedup/thinning as discussed throughout this section.

### 4.2.1 Magnitude of Melt Rates Relative to Terminus Speeds

Maximum melt rates for 2012 to 2015 are estimated to be ~8–10.5 m/d for a concentrated plume with limited spatial extent

(~100–150 m) at the terminus of Jakobshavn Isbrae (Khazendar et al., 2019), which yields a mean maximum rate of <1 m/d when averaged across the ~4.5-km width of the terminus face. Due to the non-linear relation between melt and subglacial melt discharge (Xu et al., 2013), maximum aggregate melt should be achieved when the subglacial melt emerges uniformly from beneath the terminus rather than as a discrete plume(s). Khazendar et al. (2019) indicate that their plume melt rates can be reduced by about a factor 3 to obtain the corresponding rates for uniform subglacial discharge. Since uniform rates apply to

the full terminus width, applying their scale factor to their plume rates (10.5 m/d) yields a width-averaged maximum rate of ~3.5 m/d during the recent warm ocean period. Similarly, the width-averaged maximum melt rate is ~1.9 m/d during cool periods, based on an ~5.7 m/d maximum plume rate. Note that all of these rates reflect the maximum rate at some depth, so the depth-averaged rates should be somewhat smaller (Carroll et al., 2016; Khazendar et al., 2019). It is also important to note that much of the oceanic heat in the fjord goes into melting icebergs (Moon et al., 2018), so these values may be biased high.

During the summer, the terminus advances at ~30-45 m/d, so that ice is replenished via advection far faster than it is removed via submarine melting (<1–3.5 m/d) (Joughin et al., 2012a), with calving events making up the difference.

While we cannot rule out melt serving in some way as a 'catalyst' (e.g., by undercutting the front) that accelerates calving, a decrease in average melt rate from 3.5 to 1.9 m/d over a few months of the year should not drastically slow the rate of retreat

and speedup for a glacier that moves at 30–45 m/d. For those glaciers where undercutting has been observed to have a substantial effect, the melt rate is comparable to the terminus advance rate (Luckman et al., 2015), unlike the case for Jakobshavn Isbrae where width-averaged melt rates are an order of magnitude less than terminus speeds. While a 2-D model does suggest that even modest undercutting of an idealized glacier may have some effect (O'Leary and Christoffersen, 2012), the main effect for cases near flotation is to shift a relatively weak, broad extensional stress maximum inland. A more complex

time-dependent model that includes calving with damage mechanics indicates that the effect of mélange on seasonal variation

in terminus position and speed is far greater than that of melt undercutting (Krug et al., 2015). Neither model accounts for basal crevassing, which can be important for calving near flotation (Van Der Veen, 1998). In a full 3D model that includes both basal and surface crevassing, plume melt rates of 12 m/d combined with uniformly distributed melt rates of 3.1 m/d produce little seasonally enhanced calving (Todd et al., 2018). It is only when plume melt rates are increased to ~24 m/d that there is a substantial effect for a glacier flowing more slowly (12–14 m/d) than Jakobshavn Isbrae (Todd et al., 2019). As with the 2-D model (Krug et al., 2015), the 3-D model produces a pronounced variation in terminus position and speed in response to seasonal mélange forcing, consistent with our observations.

### 4.2.2 Phasing of Submarine Melt Variability Relative to Glacier Response

In contemplating whether submarine melt, particularly in summer, might drive the observed retreat and speedup, it is important to consider the relative timing of the recent changes, particularly in relation to the season in which they occur. As Figure 6 shows, the colder water first appears in Disko Bay during the summer of 2016 (Khazendar et al., 2019). To examine whether the speeds in the summer 2016 were affected, we computed the trend (see trend in Fig. 2) for the 2012–2016 average summer speeds at M6 (see red bars Fig. 2). While the summer 2016 speeds at M6 are moderately slower than prior summers (2012–2015), this decline is consistent with the trend of declining summer speeds for 2012–2016, during which time the position of maximum summer retreat was relatively consistent. The trend of declining summer average speeds was likely a consequence of the evolving geometry that shallowed slopes and reduced driving stresses over time in the near-terminus region (Fig. 3).

The first summer when there was a substantial deviation of the speed at M6 was 2017, which was a year *after* the cold water reached Disko Bay. Some of this change is a direct response to M6 being located ~1.75 km farther upstream from the terminus from 2017 onwards. As result, the difference in speed between M6 and M9 decreased, which should have reduced summer extensional thinning in this region (the point data in Fig. 2 show some summer thinning still occurred at M6 in the summers of 2017 and 2018 before the transition to winter thickening). The reduction in speed immediately above the moving terminus (e.g., $T_{max}$-1km), however, is not as drastic as that observed at the fixed points, indicating the dominant influence of the bed depth/ice thickness near the terminus. Consequently, the difference in speed from M6 to the terminus should have produced extensional strain rates and thinning, maintaining the terminus near flotation and facilitating the observed summer calving (Amundson et al., 2010).

It is also important to note that if submarine melt were to have a significant influence on speed, it should occur through changes in the calving rate that influence terminus position, which then alter speed (e.g., Fig. 5). Since the position of minimum retreat is virtually the same in 2016 as in the prior four summers, the cooler water does not appear to have suppressed calving that summer nor did it cause the speed to deviate from the ongoing trend. Thus, the changes that led to the thickening appear to

have begun following, as opposed to coincident with, the reduction in submarine melting during the summer of 2016 (i.e., a winter-time onset).

### 4.2.3 Phasing of Mélange Variability Relative to Glacier Response

Unlike prior years in the past decade, a rigid mélange formed early in the fall of 2016 and was uninterrupted for more than 6 months. Over the ensuing winter, the terminus advanced ~6 km farther than in any winter since 2008–09 (Fig. 2), and an even greater re-advance occurred the following winter, which had similar mélange conditions. The elevation data show that the July 2016 terminus elevation was lower than any other July in the record (Fig. 3), so the advance cannot be directly attributed to a reduction in thinning that summer due to reduced melt. The additional buttressing from the partially grounded and partially floating extension that developed the following winter likely produced the much slower summer and winter speeds, which in turn produced the observed thickening thereafter (Fig. 3; see also Khazendar et al. 2019). The winter of 2018–19 also had a more extended than normal period of rigid mélange, but less so than the previous two winters. In this instance, the winter advance fell between the extremes of 2011–2016 and 2016–2018 periods, with a moderate increase in the summer 2019 speeds relative to 2018. Collectively, these observations are consistent with earlier work indicating that the presence of rigid mélange can suppress calving on Jakobshavn Isbrae (Amundson et al., 2010; Joughin et al., 2008b).

Consistent with our observations for Jakobshavn Isbrae, mélange also appears to affect terminus advance and retreat on Helheim and Kangerlussuaq glaciers (Kehrl et al., 2017). For Kangerlussuaq Glacier, an observed reduction in mélange rigidity during the winters of 2016–17 and 2017–18 appears to have produced a substantial retreat and speedup (Bevan et al., 2019). Moreover, full Stokes model simulations that include the influence of mélange produce seasonal variations that are consistent with those we observe for Jakobshavn Isbrae (Todd et al., 2018; 2019).

Time series of Sentinel 1A/B SAR imagery from 2015-2016 reveal regular flushing of the mélange from the fjord (see movie in Supplement) throughout the summer, consistent with the lack of rigid mélange at the terminus (Fig. 2). While not as regular as in the summer, in the winter of 2015–16 there were several flushing events, during which a large portion of the mélange was rapidly advected through and cleared from the fjord. In the 2016–17 and 2017-18 cold seasons, however, the first flushing events did not occur until April and May, respectively, consistent with the indication of more rigid mélange at the times shown in Fig. 2.

### 4.2.4 Mélange Rigidity and Relation to Atmosphere and Ocean Forcing

It is well established that rigid mélange is most prevalent during winter when surface air temperatures are low (Cassotto et al., 2015), which agrees with the results in Fig. 2. Nearby weather station data (Fig. 6) indicate that the 2016–17 and 2017–18 winters were moderately colder than normal for the decade, but so was the 2014–15 winter that was followed by a large summer speedup, suggesting that air temperatures are not the only control on mélange rigidity. As noted above, 250-m water

temperatures in Disko Bay were ~1.5°C cooler for 2016–2017 (Fig. 6 and Khazendar et al., 2019), and this cold ocean layer should extend across the sill at the mouth of the fjord (Gladish et al., 2015a). A reasonable assumption is that this colder water facilitated the more rapid winter freeze up and greater mélange rigidity throughout the winter of 2016–17, with similar behavior during subsequent winters. By contrast, the warmer layer of water that was present from 2011–2015 likely facilitated greater mélange mobility, which allowed greater calving throughout the corresponding winters. This hypothesis is consistent with the finding that much of the oceanic heat in a mélange-choked fjord contributes to iceberg melting (Moon et al., 2018).

The 2010–11 winter is interesting because the amplitude of the annual cycle of terminus advance and retreat was reduced relative to other winters (Fig. 2). Periods of rigid mélange were sporadic, but occurred relatively early in the autumn and relatively late in the spring. Ocean temperatures were also exceptionally cool in 2010 (Fig. 6 and Gladish et al., 2015b), which may have contributed to the formation of the more rigid mélange that was present at times during the late spring and early fall. By contrast, the 2010 and 2011 winter air temperatures preceding and following summer 2010 were the warmest since before 1980 (Fig. 6), which may have reduced the formation of rigid mélange in winter. Thus, both oceanic and atmospheric forcing, acting counter to the 'normal' seasonal cycle, may have played a role in the unusual retreat pattern for 2010. Similarly, but to a lesser extent, the slight reduction in mélange rigidity for the 2018–2019 winter relative to the two prior winters may be the result of some combination of slightly warmer summer ocean and winter air temperatures (see Fig. 6). Thus, while the influence of ocean temperature may be the dominant factor in the formation of rigid mélange, air temperature may play a secondary role in determining interannual variability of mélange rigidity. In summary, our observations suggest that while recent fluctuations in ocean temperature over the last decade may have governed much of the flow variability of Jakobshavn Isbrae (Khazendar et al., 2019), the primary link between ice flow speed and ocean forcing is through the influence of water temperature on mélange rigidity, not submarine melting at the terminus.

### 4.2.5 Relation of Melt and Mélange Variability to the Late 1990s Ice Tongue Breakup

The data in Fig. 6 show both cooler ocean and atmospheric temperatures prior to the late 1990s speedup, consistent with higher sea-ice concentration in Disko Bay during this period (Joughin et al., 2008b). The original breakup of the floating ice tongue in the late 1990s early 2000s and the subsequent speedup has been attributed both to reduced mélange rigidity (Joughin et al., 2008b) and to greater submarine melting (Holland et al., 2008; Motyka et al., 2011). Unlike the configuration in recent years when the terminus often had a vertical face with limited area in contact with the ocean, the former 15-km long, several hundred-meter-thick floating tongue had an order of magnitude or more greater area that was exposed to high melt. Indeed, earlier work noted that a basal melt rate increase of 1 m/d or more could have had a strong effect on the ice tongue viability (Holland et al., 2008; Motyka et al., 2011). From 1962 to 1996, however, there was a strong seasonal variation in calving likely driven, at least in part, by mélange rigidity (Sohn et al., 1998), which could have been amplified as ocean and air temperatures increased. While submarine melting may have thinned the ice shelf substantially, thin ice can remain intact for long periods when embedded in perennial sea ice or strong mélange (Reeh et al., 2001). Since there likely were occurrences of both greater melt

and reduced mélange rigidity over the preceding several years, it may have been their combined effects that eliminated the ice tongue and triggered the late 1990s speedup. In addition, enhanced hydrofracturing due to increased surface meltwater may have played a role in breaking up the ice tongue (Scambos et al., 2000; Sohn et al., 1998).

### 4.3 Transverse Depressions Leading to Calving Events

Beginning in 2012 and extending through at least 2015, a series of transverse depressions developed a few kilometres upstream of the terminus prior to several late summer and autumn calving events (Fig. 3). The bottoms of some of these depressions extend more than 50 m below flotation, and appear to eventually serve as the detachment points for large calving events. The surface between the terminus and these depressions can reach heights of 25 m or more above flotation. The development of one on Jakobshavn Isbrae was captured with a terrestrial radar interferometer in August 2012 (Cassotto et al., 2019), and similar features have been observed on Helheim and Kangerdlugssuaq glaciers (Howat et al., 2007; James et al., 2014; Joughin et al., 2008c). These features, which develop over weeks, are distinctly different than the depressions that can form as the terminus rises above flotation for tens of minutes prior to large calving events (Parizek et al., 2019).

To better examine the evolution of one of these depressions in 2015, which was most prominent in our record on September 30, Figs. 7 and 8 show a time series of elevations in profile- and map-view, respectively. To improve temporal sampling, Fig. 8 also shows several TerraSAR-X images. The imaging geometry for these acquisitions was such that a prominent radar shadow is visible when there is a steep, high terminus, (e.g., August 24). When they are well-developed, depressions are often distinguishable in the SAR data (e.g. September 26). The general pattern of brightening with time in the SAR image time series is due to the transition from summer melting to fall freeze-up. To track the evolution of the parcel of ice that evolves to form the bottom of the large depression, we used the average velocity field for this period to estimate its location through time, which is shown with coloured triangles in Fig. 7 and white dots in Fig. 8.

Prior to the formation of the depression we examine here, in the July 31 and August 5 DEMs, a minor depression near flotation is apparent (Fig. 7), which is difficult to identify in the SAR images from this period (Fig. 8). These first two DEMs and the SAR data reveal a nearly vertical calving face ~120–140 m high. By August 13, the less pronounced shadow in the SAR image indicates a less sheer terminus and the development of several large crevasses or, potentially, through-going rifts. By August 22 a large calving event, which appears to have extended through the bottom of the minor depression, produced a steep new calving face ~130 m high. Although the August 22 DEM is limited in extent, there is a shallow back slope at the upstream limit of the DEM extent that suggests the presence of the nascent depression, which is not yet distinguishable in the radar image from 2 days later. By September 4, however, there is a subtle indication of the depression in the radar data (see area around white circle for September 4 in Fig. 8). The September 14 elevation data show that the depression evolved substantially, with its low point then ~40-m below flotation (Figs. 7&8). Over the next 16 days, this feature advected downstream, deepening to ~65 meters below flotation. Although the spatial scale is such that the area below flotation is not fully in hydrostatic

equilibrium, its depth suggests thinning by up to several hundred meters. Sometime before October 11, a large calving event(s) removed area seaward from the bottom of the depression.

Depressions below flotation that develop through-going rifts and form detachment points for icebergs represent one mode of
calving (James et al., 2014; Joughin et al., 2008c). One mechanism that has been proposed to explain this evolution involves a grounded glacier advancing down a prograde (forward) slope, driving the terminus downward, causing buoyant flexure that lifts the front above flotation but depresses the region upstream below flotation (James et al., 2014; Wagner et al., 2016). This process does not seem to apply here because both the available bed topography and the observed speed as function of terminus position (e.g., Fig. 5) are consistent with a retrograde rather than prograde bed slope. Moreover, the glacier surface is above
flotation both upstream and downstream of the depression, which means if it was purely the result of downward flexure of ice initially above flotation then its base would have to dip below the bed. Alternatively, flow models illustrate that reverse surface slopes can develop near the terminus of an outlet glacier, but these features are generally tied to the underlying topography and do not advect downstream as do the transverse depressions shown in Figs. 7 and 8 (Vieli et al., 2002).

In general, the transverse depressions we observe tend to develop after summer thinning yields a surface near flotation in the region a few kilometres upstream of the terminus (Fig. 3). Strong extension near the terminus and an ice column that is at or near flotation should facilitate the development of basal crevasses (Van Der Veen, 1998). Once a basal crevasse develops, the thinner ice above it must support the same column-integrated longitudinal stress, leading to greater extension. As a result, a 'necking' process may commence, causing sustained localized thinning that would create depressions like those shown in Figs.
3, 7, and 8 (Bassis and Ma, 2015). Consistent with this hypothesis, the observed depressions have width/thickness ratios (~1:1) similar to modelled results (Bassis and Ma, 2015). Furthermore, Fig. 7 shows the velocity gradient steepening over the areas where the depressions occur, which is consistent with that increased strain rates and localized thinning that forms them.

As the depressions develop, the elevation downstream of the depression and ~1 km upstream of the terminus remains tens of
meters above flotation (Figs. 7 and 8). While similar high spots have been attributed to buoyant flexure on Helheim (James et al., 2014), in the examples presented here, the downstream high spot seems to be advected downstream faster than it can thin to flotation (see progression of high spot from August 22 to September 30). As this region increases its elevation above flotation, strong extension produces thinning so that the last few hundred meters of the heavily rifted terminus regions go afloat.

The development of other depressions like those just described occurred over several weeks, culminating in large calving events involving a kilometre or more of ice (e.g., August 22 to October 11 in Figs. 7 and 8). Such events seem to occur in late summer to early fall and did not start until 2012. As Fig. 3 indicates, in 2010 and 2011 the near-terminus region was relatively steep, rising quickly above flotation, making it less likely that basal crevasses will form upstream of the terminus at these

475 times. The more intense summer speedups that began in 2012, however, produced shallower late-summer calving fronts, with elevations near flotation extending several kilometres upstream of the terminus, which were potentially more favourable to basal crevasse formation that could seed the formation of these depressions. Thus, this mode of calving appears generally to occur in late summer after strong speedups have produced conditions favourable to the formation of basal crevasses that can evolve to form surface depressions. Since full-thickness calving requires a terminus near flotation (Amundson et al., 2010),

the development of the high spots well above flotation downstream of the depressions may suppress calving long enough to give these features more time to develop.

## 4.4 Terminus Elevation and Cliff Instability

The depressions that lead to the calving events described above represent just one of a variety of failure modes that can cause calving. Recent work has raised concern about ice-cliff instability contributing to rapid ice-sheet collapse (DeConto and Pollard, 2016).

Pollard, 2016). With ice cliffs more than 130 m above the waterline at times (see Fig 7), Jakobshavn Isbrae provides an interesting case study for understanding the effect of such potential instabilities. In actuality, however, ice sheet or glacier instability arises when the calving and discharge rates fall out of sync (Amundson and Truffer, 2010), rather from the actual failures themselves (i.e., calving events). Whether it initiates above or below the water line, it is important to note that some kind of material failure must occur to produce calving events even in steady state. It is also important to consider that while

the Jakobshavn terminus is advancing at ~40 m/d and typically calving in ~100 m slabs, any ice cliff has a limited lifespan (~2.5 days on average, though clustering of calving events may keep some cliffs intact up to weeks; see Fig. 7). Here we examine evolution of the front, following the formation of steep calving faces.

Figure 7 illustrates the evolution of what begins as an initially ~130-m high sheer ice front (see August 22 in Fig. 7). Rather

than a brittle failure event leading to a rapid collapse, the strong extension near the terminus instead causes evolution from a sheer grounded cliff to a heavily crevassed floating tongue with a ~5% slope (see September 30 in Fig. 7). In some sense this progression represents the failure of the cliff, but as a process that evolves over weeks likely involving multiple small failures (e.g., crevasse events) rather than a single catastrophic failure. Furthermore, the stability of the floating extension depends on environmental factors. As our data suggest, cooler ocean temperatures, perhaps supplemented by colder air temperatures, could

allow such a tongue to advance over the colder part of the year. Conversely, when more summer-like conditions prevail, such a tongue should disintegrate far more rapidly as was the case for the transient tongue shown in shown in Figs. 7 and 8. In this case, in addition to the lack of rigid mélange, submarine melt may contribute to the breakup by thinning the floating section at rates of up to a few m/d. Thus, stability likely is governed more by environmental conditions than by a single height-dependent mechanical failure criterion that could yield rapid collapse (Parizek et al., 2019).

Our data also reveal that once the summer speedup commences, near terminus surface thinning rates can exceed 50 m over the course of a few months (see M6 elevations in Fig. 2). Thus, even without cliff failure, the high stretching rates associated with

an unbuttressed terminus on a retrograde bed slope can cause rapid thinning to flotation, which, if unabated, will lead to further calving and rapid retreat. Were it not for the seasonal cycle that produces re-advance and winter thickening (see Figs. 2 and 3), the terminus of Jakobshavn Isbrae would have receded far deeper inland than it has thus far, even without ice cliff collapse as such.

Finally, it is important to note that to the best of our knowledge, Jakobshavn Isbrae has the deepest unbuttressed calving face in Greenland or Antarctica, leading to the fastest marine-terminating glaciers speeds (Joughin et al., 2014). As such, it has been out of balance at times by more than a factor of 3 seasonally (Joughin et al., 2014) and a factor of 2 annually (Mouginot et al., 2019). Nearly all glaciers in Greenland and Antarctica have steady-state speeds well below that of Jakobshavn Isbrae (Joughin et al., 2010; Rignot et al., 2011). Thus, given that speed scales non-linearly with terminus depth (Schoof, 2007), any glacier that evolves to the point where the height of its unbuttressed calving face rivals that of Jakobshavn Isbrae will already be well out of balance, yet likely will maintain heights well below those needed to exceed a material failure criterion (~200 m) that would lead to rapid brittle failure (Parizek et al., 2019). Our point is not that cliff failure is irrelevant, but rather that any glacier that reaches terminus heights where cliff failure may be a significant effect, must already be in a state of rapid retreat. Far more important to long-term outlet glacier stability is how seasonal variability and oceanographic/atmospheric trends influence the calving rates of grounded glacier termini. In Antarctica and northern Greenland there are additional atmospheric temperature sensitivities (e.g., meltwater ponding and hydrofracture) that influence ice-shelf stability (Scambos et al., 2000). In summary, while brittle failure cliff instability may be important in some circumstances, it is far more likely to play a role in a late stage retreat than to serve as the process that would initiate such a retreat.

## 5 Conclusions

We have assembled and produced a comprehensive time series of terminus position, surface flow velocity, surface elevation, and mélange rigidity for Jakobshavn Isbrae over the last decade. The data show a strong degree of variability, including a potentially brief (a few years) slowdown that coincided with cooler ocean temperatures (see also Khazendar et al., 2019). The time series of elevation provides an unprecedented level of detail, which clearly shows a pattern of summer thinning partially offset by winter thickening in response to seasonal changes in flow speed over most of the record. At least from fall 2016 through spring 2019, winter thickening outpaced summer thinning, leading to net thickening and elevations approaching those observed in 2010. These data also provide observational evidence to support theoretical development describing how necking proceeds as basal crevasses form (Bassis and Ma, 2015). The elevation data also show that although Jakobshavn Isbrae likely has the highest unbuttressed ice cliffs on Earth, at this point they do not appear to be subject to sustained catastrophic brittle failure. Most importantly, our observations reinforce earlier findings on the influence of mélange rigidity on calving (Amundson et al., 2010; Joughin et al., 2008b; Krug et al., 2015; Todd et al., 2018), and help establish an apparent connection to ocean temperature. Ocean temperatures are expected to rise over the next century (Stocker et al., 2013), which will likely

produce further retreat of Jakobshavn Isbrae.  Superimposed on any trend for the last century, however, there is substantial multi-decadal scale variability of ocean temperatures in Disko Bay that correlates well with the Atlantic Multidecadal Oscillation (AMO) Index, which has been linked to past changes on Jakobshavn Isbrae (Lloyd et al., 2011). Thus, whether Jakobshavn Isbrae can stabilize, at least temporarily, likely depends on whether a cycle similar to that of the last century produces an extended period (several more years to decades) of cooler waters in Disko Bay. While our results should be

applicable to glaciers with high calving rates that yield a thick mélange (Bevan et al., 2019; Kehrl et al., 2017), more work is needed to understand the influence of thinner mélange on smaller glaciers that calve less rapidly.

**Data Availability**

The pre-2019 velocity data are available at NSIDC (NSIDC-0481 at  https://nsidc.org/data/measures/gimp) and the 2019 data will be archived there by the end of 2019. The elevation, terminus position, and mélange data are staged at the UW library

(**archiving in process doi will be provided here in the final copy**).

**Author Contribution**

All co-authors contributed to the production of the data sets:  DEMS (DS, DF, and BS), velocity (IJ), and terminus position (IJ). IJ prepared the manuscript with contributions to the analysis and discussion from all co-authors.

**Acknowledgements**

The National Aeronautics and Space Administration (NASA) supported contributions by I.J. (NNX17AH04G), D.S. (NESSF fellowship award NNX12AN36H), and B.S. (NNX13AP96G). Resources supporting this work were provided by the NASA High-End Computing (HEC) Program through the NASA Advanced Supercomputing (NAS) Division at Ames Research Center. The TerraSAR-X (L1b SSC products) and TanDEM-X (L1b CoSSC products) data were provided by DLR (project HYD0754 and XTI_GLAC0400) through their German Remote Sensing Data Center (DFD) and are subject to DLR copyright.

The Cosmo-SkyMed data were delivered under an ASI licence to use. The bed data were provided by the BedMachine project at UC Irvine and archived at NSIDC (https://nsidc.org/data/IDBMG4). Ocean temperature date were provided by the Ocean Melting Greenland Project (OMG) and the International Council for the Exploration of the Sea Oceanography. Weather station data were provided by the Goddard Institute for Space Studies GISTEMP Team. We acknowledge Claire Porter, Paul Morin, and others at the Polar Geospatial Center (NSF ANT-1043681) who managed tasking, ordering, and distribution of the L1B

commercial stereo imagery under the NGA NextView license. We also acknowledge Delwyn Moller and the OMG team for the GLISTIN data. DEMs provided by the Polar Geospatial Center under NSF-OPP awards 1043681, 1559691, and 1542736. Comments by M. Maki, A. Luckman, A. Khazendar, J. Willis, I. Fenty and the anonymous reviewer improved the manuscript.

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

**Figure Captions**

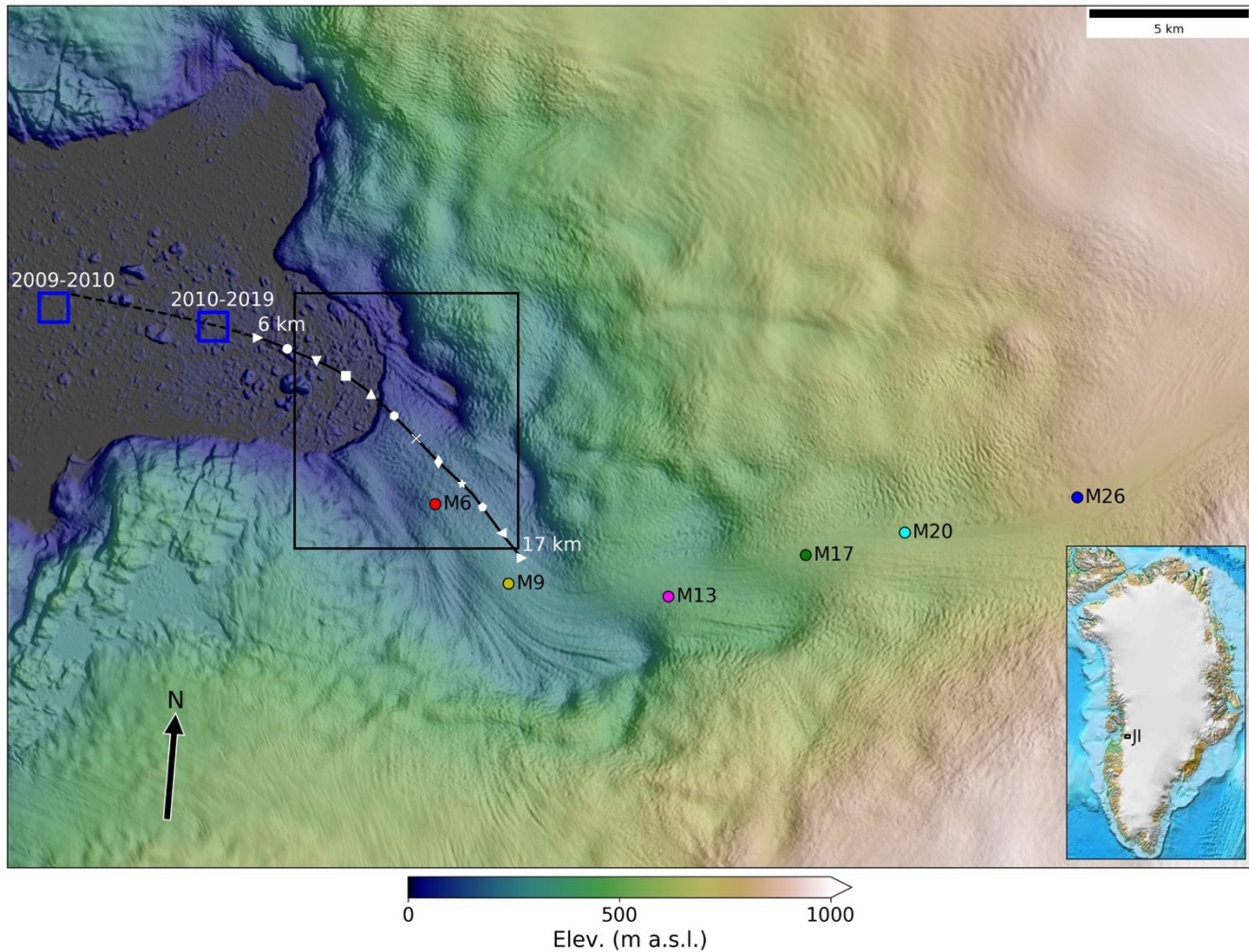

Figure 1: Surface elevation (colour) over shaded relief map derived from the Greenland Ice Mapping Project (GIMP) DEM for the region near the terminus of Jakobshavn Isbrae (see inset for location). The black box shows the location of the images shown in Figs. 4 and 8 and the blue box indicates the areas used to sample mélange as described in the text. The solid black line shows the location of the profile shown in Figs. 3 and 7, with the dashed section used to show the full extent. White markers with assorted 730 symbols denote 1-km intervals along the profile between 6- and 17-km. Coloured markers (M6-M26) show where speed is sampled in Fig. 2 at points consistent with earlier work (Joughin et al., 2008b; 2012b; 2014).

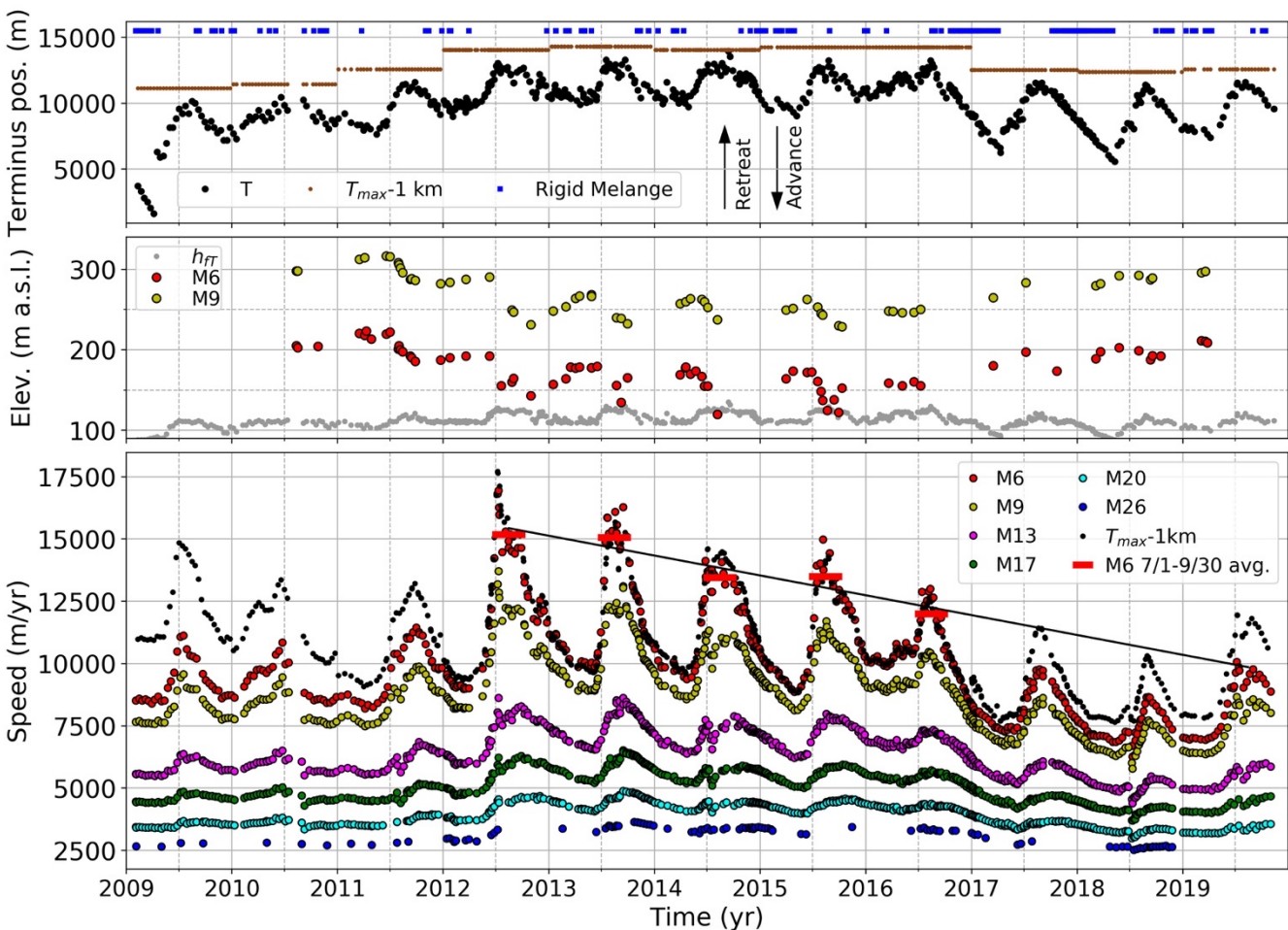

**Figure 2. (top)** Terminus position (T: black circles), and position 1-km upstream of annual minimum terminus extent ($T_{max} - 1$km: brown dots). Terminus positions are given as distances measured relative to the seaward origin of our reference profile (see Fig. 1) and the arrows show direction of retreat and advance. Two points in 2014 were excluded in determining $T_{max} - 1$km (see text). The blue squares indicate when rigid mélange was present in front of the terminus at the locations indicated by the blue boxes in Fig. 1. **(middle)** Elevations of the points M6 (red) and M9 (gold) extracted from all available DEMs along with the inferred flotation height, $h_{fT}$, (grey) at the terminus. **(bottom)** Surface speeds through time extracted from a TerraSAR-X/TanDEM-X velocity time series, with a few points from summer 2019 derived using COSMO-SkyMed data. See Fig. 1 for the locations of points M6-M26. The black circles indicate the speed at $T_{max} - 1$km. Because this minimum position is updated annually, unlike the static points, there are discontinuities at each year boundary since the sampling point location changes (e.g., brown points in the top plot). The average summer (June 1 to September 30) speeds are shown as red bars for the five summers when the terminus was most retreated (2012–2016). A linear fit ($r^2$=0.01, p=0.013) to these summer averages is shown with a black line that extrapolates the trend through 2019.

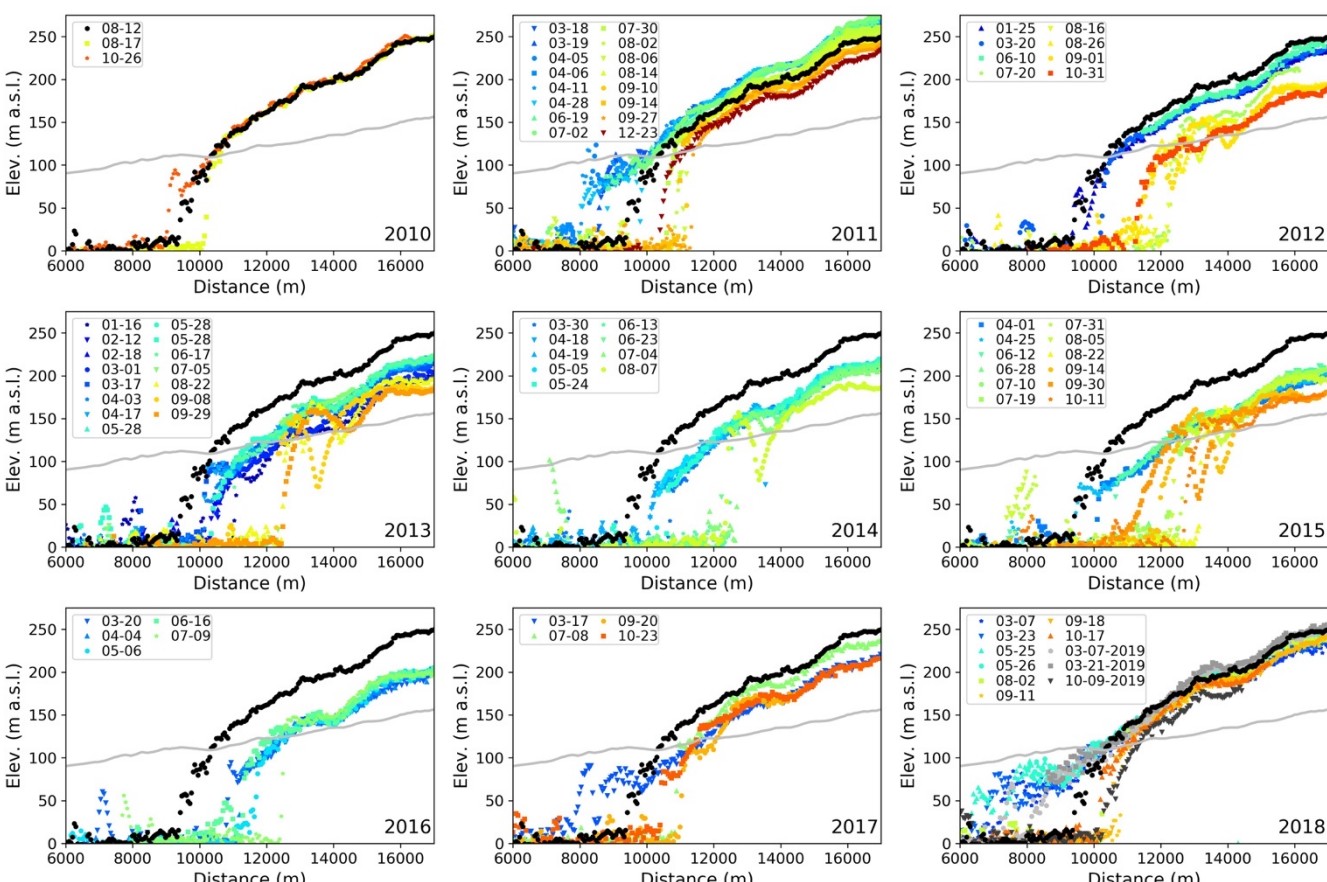

Figure 3. Plots for each year from 2010 through 2018 showing the available near-terminus elevation profiles, with three 2019 profiles included in the 2018 plot (see Fig. S2 for separate 2018 and 2019 plots). Profiles are labelled by date (MM-DD) and the profile location is shown in Fig. 1. For reference, the initial 2010 profile (2010-08-12) is included in each plot. All elevations are relative to nominal mean sea-level (EGM2008 geoid). The grey line indicates the flotation height as estimated using bed elevations (Fig. S1) from BedMachine v3 (Morlighem et al., 2017).

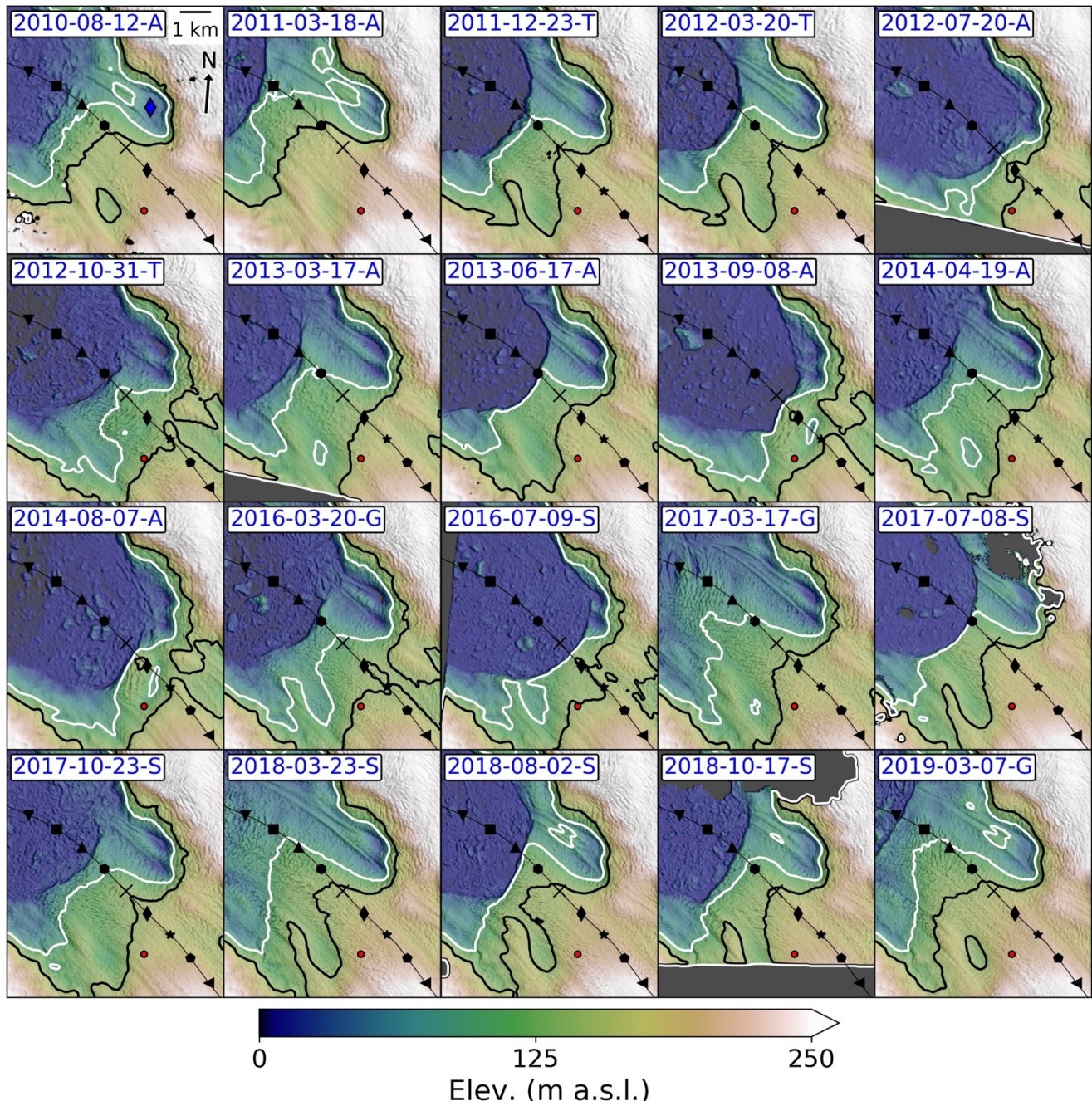

**Figure 4.** Colour shaded relief maps for of a subset of DEMs, excluding maps for 2015, which are included in Fig. 8. Elevation contours of 150 (black) and 100 (white) meters bound the approximate transition from grounded to floating ice. The elevation data were lightly smoothed prior to contouring to eliminate crevasse noise. The letter after the date-string (YYYY-MM-DD) indicates the DEMs source (A: ASP World View, T: TanDEM-X, S: SETSM World View, and G: GLISTIN). The profile with symbols every 1-km is the same as shown in Fig. 1 and the red dot shows the location of M6. The blue diamond in the 2010-08-12 panel is referenced in the text.

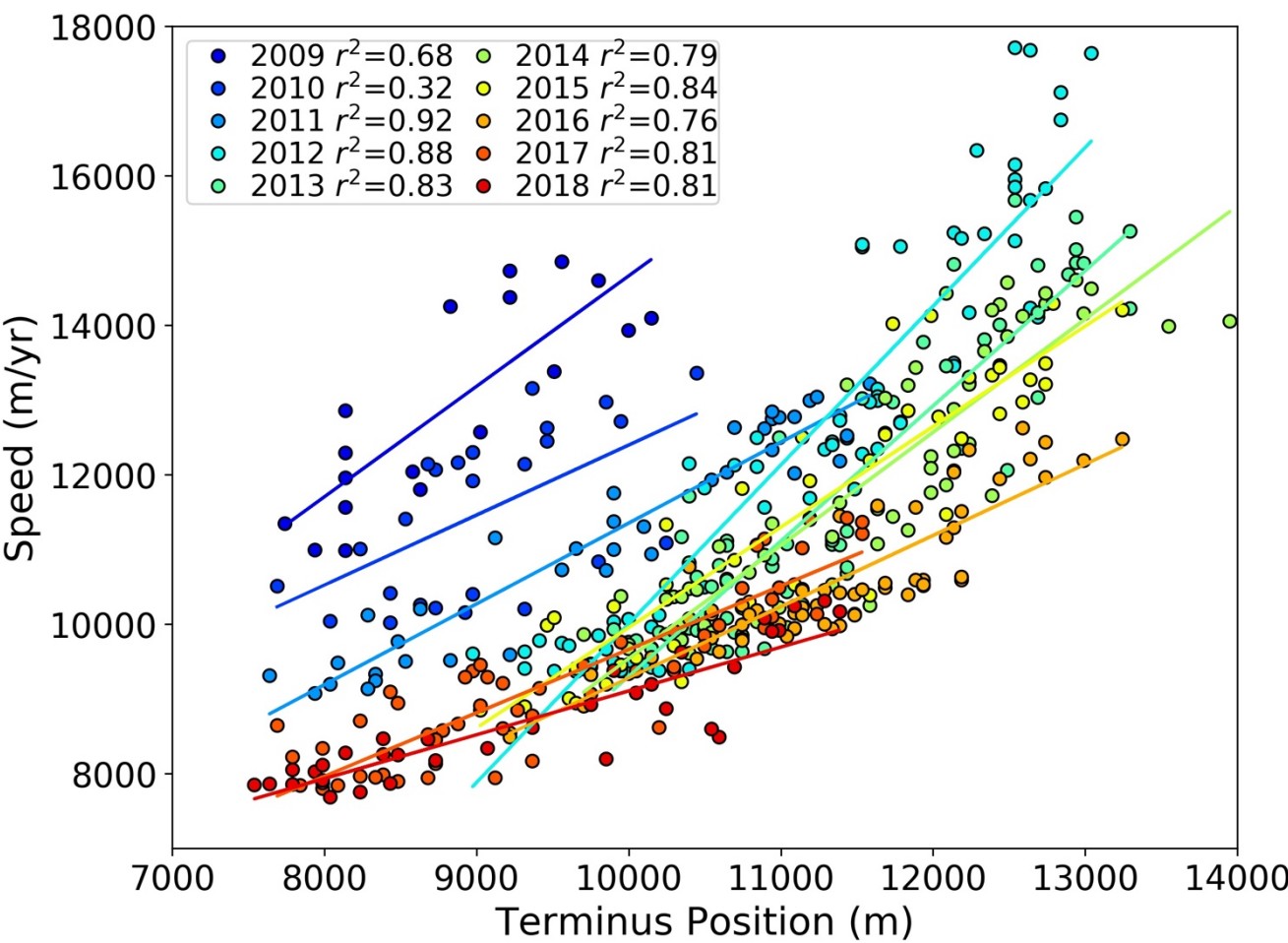

**Figure 5. Speed at T$_{max}$-1 km (see grey points Fig. 1) as function of terminus position (relative to the profile shown in Fig. 1) for each year from 2009 to 2018, with greater terminus positions indicating greater retreat. Lines show linear regressions for each year with r² values included in the legend.**

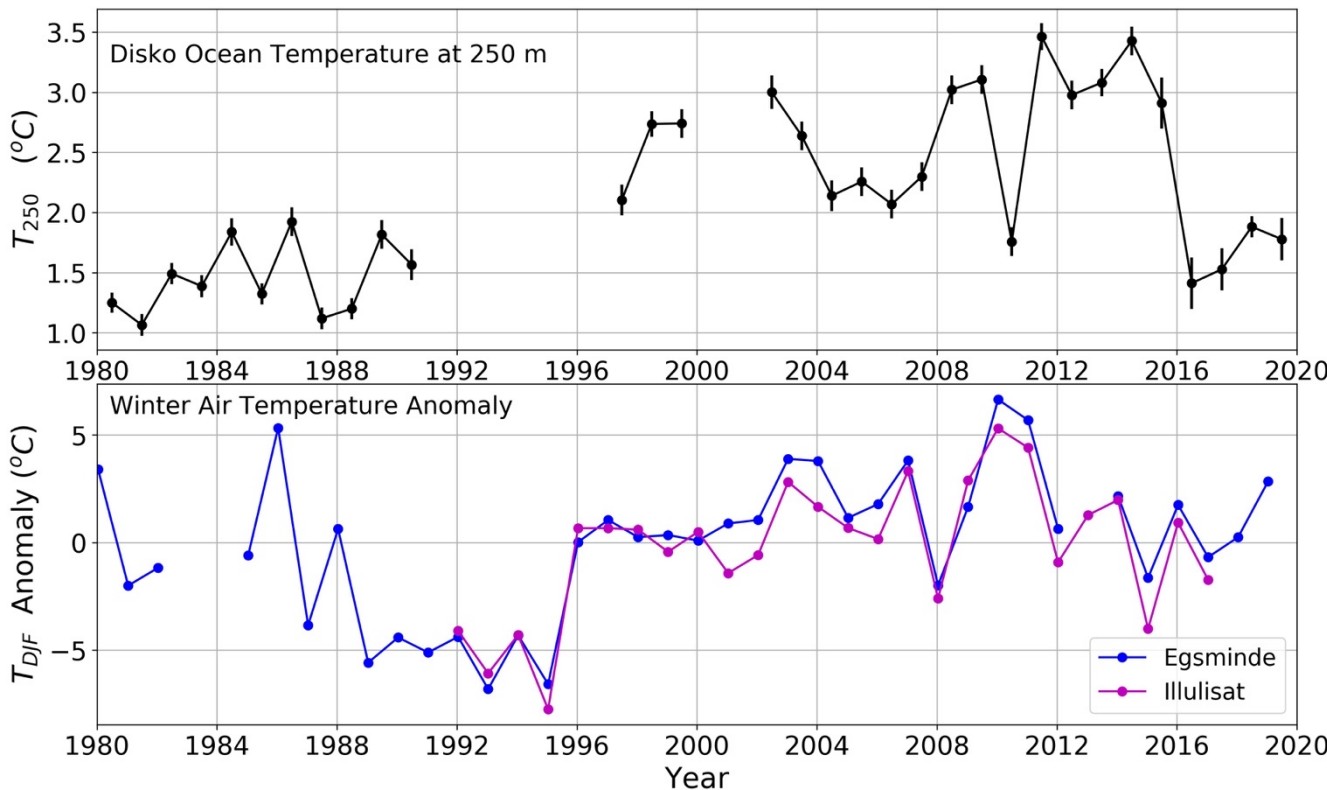

**Figure 6. (top)** Ocean temperatures at 250-m depth in Disko Bay (all available casts in the region: 68.6° < latitude < 69.4°, -54° < longitude < 52°), which is based on a similar figure by Khazendar et al. (2019). Small differences with their results exist based on the sampling region and the averaging scheme. The ocean temperatures for 2015–2019 are extracted from the Ocean Melting Greenland data set (https://doi.org/10.5067/OMGEV-AXCTD) and for 1980–2014 are extracted from the International Council for the Exploration of the Sea Oceanography (http://www.ices.dk/marine-data/data-portals/Pages/ocean.aspx) database. To obtain estimates representative of 250-m depth, we averaged temperatures from 225 to 275 m depth. The bars indicate uncertainty in the mean based on the number of points in the sampling region. To compute them, we take the mean intra-annual standard deviation averaged over the full record as the intrinsic uncertainty for a single point (i.e., how representative a point measurement is for the entire region). We then scale this value by 1/sqrt(number of points) to determine the plotted bar for each annual mean. **(bottom)** Winter 2-m air temperature (D-J-F) anomalies relative to the mean for the period from weather stations in Egsminde and Illulisat (GISTEMP Team 2019, 2019).

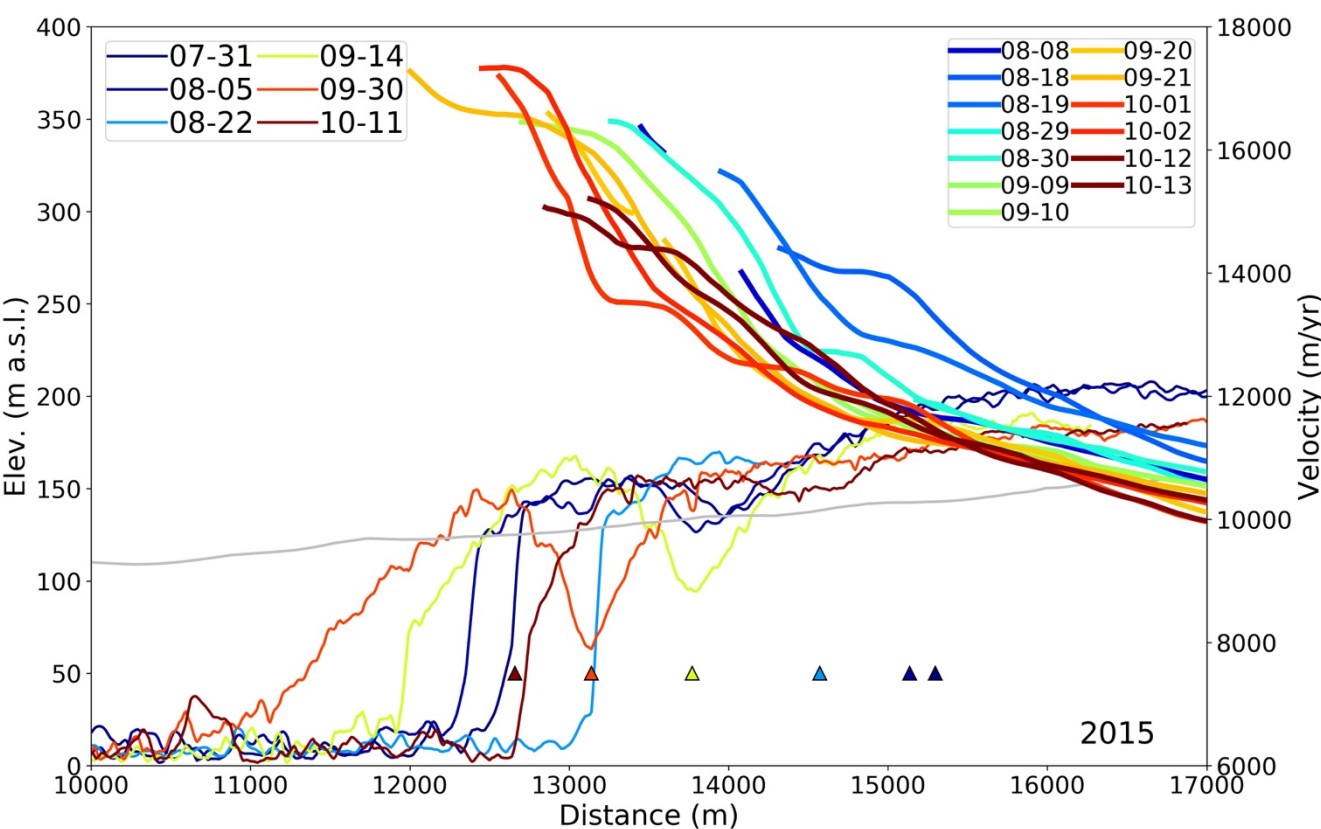

Figure 7. Elevation and velocity profiles for summer 2015 illustrating the evolution of an advecting transverse surface depression. The legends give the nominal dates (MM-DD) in 2015. As in Fig. 3, the grey line indicates nominal height of flotation. The triangles show the estimated locations (see text) of parcel of ice that evolved to form the bottom of the depression discussed in the text (see also white circles in Fig. 8), which served as a detachment point for a calving event(s) that occurred between September 30 and October 11.

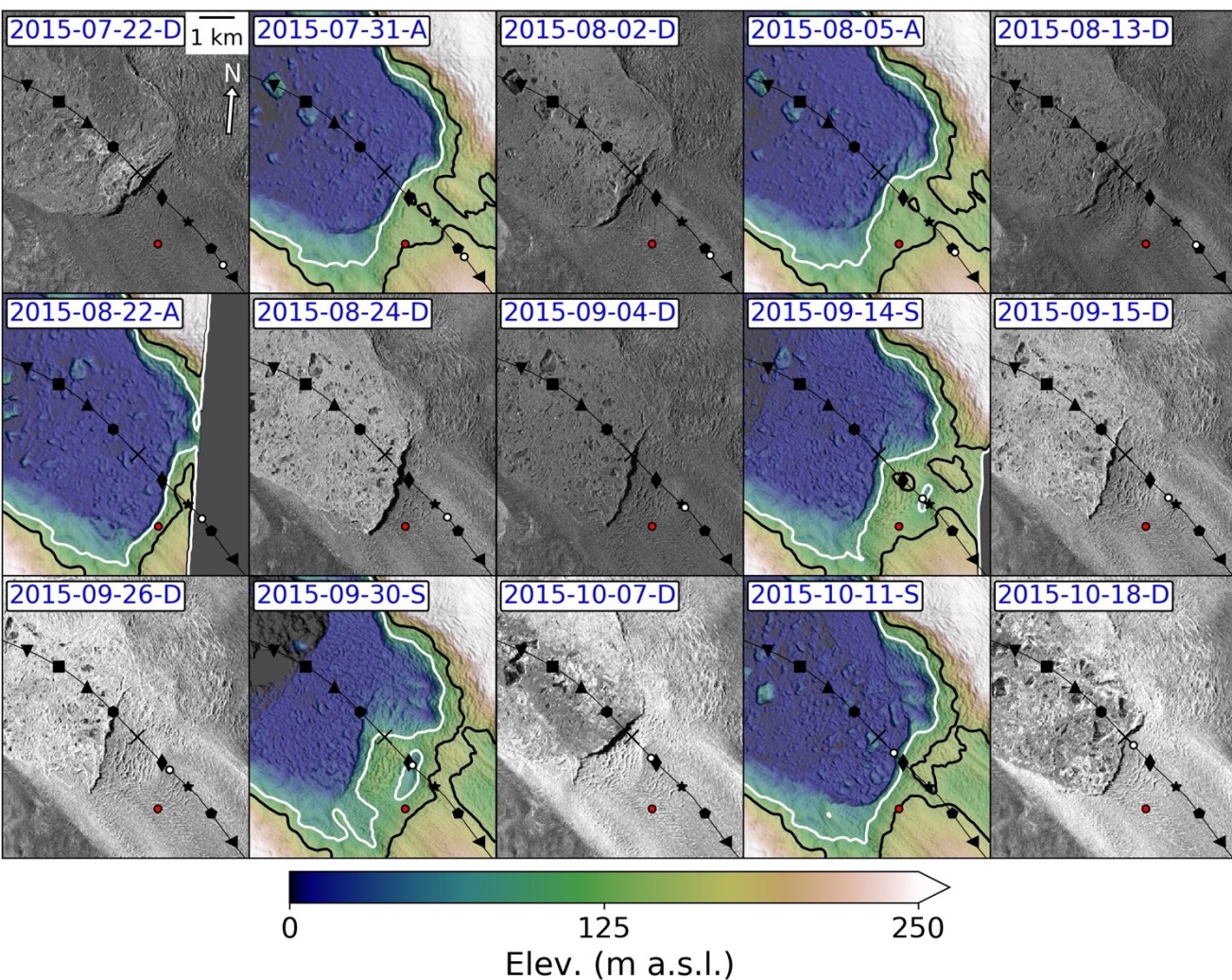

**Figure 8. Time series of TerraSAR-X SAR backscatter amplitude imagery and colour shaded relief maps for late summer 2015 identified by date of acquisition (YYYY-MM-DD). The source of the data in each map is indicated by the letter after each date (D: TerraSAR-X images copyright DLR 2015, A: ASP World View DEM, S: SETSM World View DEM). The artificial illumination source for the DEM shaded surfaces was chosen to approximate the illumination angle of the TerraSAR-X radar. Note that the terminus in the radar imagery casts a true shadow, while the corresponding shadows in the shaded relief maps are based on the local slope/aspect. The red circle shows the location of M6, while the white circle tracks the parcel of ice that evolved to form the depression (e.g. September 30) described in the text (see also triangles Fig. 7).**