# Peer review of "A Decade of Variability on Jakobshavn Isbrae: Ocean Temperatures Pace Speed Through Influence on Mélange Rigidity"

_The Cryosphere, 2019_

## Short Comment (SC1) · 23 Sep 2019

This Short Comment focuses exclusively on correcting some inaccurate representations of our findings that we reported in Khazendar et al. (2019). We gratefully thank the authors in advance for their kind consideration.

- The authors incorrectly characterize one of the main conclusions of our study in Lines 276-282: "Maximum melt rates for 2012 to 2015 are estimated to be ∼8-10 m/yr for concentrated plumes with limited spatial extent, or about a factor 3 less if the melt water emerges with a uniform distribution from beneath the grounded ice (Khazendar et al., 2019). Whether concentrated or evenly distributed, the factor-of-3 reduction

should roughly represent the average melt rate across the terminus. Thus, scaling the plume rates from Khazendar et al. (2019) by a factor of 3 yields approximate average melt between ∼3.5 m/yr during the summers with warmest water and ∼1.9 m/yr during the summers when the water was coolest."

In their argument, the authors take our values of maximum melt rates and translate them to lower mean rates. Yet, we opted to present the maximum melting rates purposefully. For deep glaciers such as Jakobshavn Isbrae, ocean-induced melting at the front tends to reach its maximum value within ∼100 m of the grounding line up the face of the glacier (Carroll et al., 2016). This enhanced melting has been observed to produce widespread undercutting of the glacier fronts (Fried et al., 2015; Rignot et al., 2015), which could lead to increased calving, frontal retreat and reduced resistance to flow. Observations and theoretical work have suggested that calving can be a direct response to undercutting at the front (Bartholomaus et al., 2013; Luckman et al., 2015) and that submarine melting and undercutting can contribute to calving that is several times the melting rate (O'Leary and Christoffersen, 2013; Benn et al., 2017; Todd et al., 2018). Therefore, rather than scaling our quoted melt rates down, it is more likely that the melt rates should be scaled up to represent their potential impact on the calving rate.

Furthermore, we emphasized in our paper that while we found that the glacier's thickness changes had a strong correlation with ocean temperature variability, the former was even more strongly correlated with the variability in submarine melting rates.

- The authors then continue on Lines 289-291 with another argument to justify rejecting the relevance of submarine melting: "Although submarine melt should have been substantially reduced in the summer of 2016 (Khazendar et al., 2019), the maximum retreat was virtually identical to that of the four prior years and speeds were only slightly reduced, suggesting melt is not directly controlling retreat."

This is a mischaracterization of our data and approach.

It is clear from the Davis Strait mooring data (Figure 3 in Khazendar et al., 2019) that cold water started arriving in Disko Bay in June or July of 2016, and in fact Figure 2 of Joughin et al. shows that glacier speeds in 2016 at Tmax-1km, M6 and M9 all experienced the smallest increases between the spring minimum and the summer peak than any other summer since 2011. Our Figure 3 shows the same.

As we state in our paper, the ocean properties and subglacial discharge volumes we use in calculating submarine melting rates are from the summer of each year, while the thickness changes of Jakobshavn are from the following spring, when the altimetry data were acquired. Regarding the year 2016, this is what we stated: "Most prominently, the sharp drop in ocean temperatures in 2016 and 2017 by 2 °C relative to the peak temperature in 2014 corresponds to the slowing and dramatic thickening of the glacier in 2017 and 2018." Indeed, our observations (Fig. 3 in Khazendar et al., 2019) show that the flow speed of Jakobshavn starts a significant slowdown in the summer of 2016, around the time of the observation of the large drop in ocean temperatures and submarine melting rates. The glacier then reaches its slowest flow speeds in the spring of 2017, coinciding with our measurement of significant thickening. The flow speed then stages only a weak recovery in the summer of 2017. This pattern is also shown by the data in Figure 2 of Joughin et al.

Our rendering of the events and their relative timing is consistent, so we request that authors remove the text on Lines 289-291 in its current form as it misconstrues our findings.

More generally, we aimed to be careful in framing the conclusions of our study as not to claim that ocean temperature variability and submarine melting are the sole explanations of Jakobshavn's dynamic evolution. We wrote that we "find the evidence sufficient to conclude that ocean temperature variability, through its influence on submarine melting rates, has been a main, and sometimes dominant, factor in shaping Jakobshavn Isbrae's interannual dynamic evolution since the disintegration of the ice shelf in 2003." We feel this conclusion holds without us having to dismiss the possibility that other

processes might also have had a role in shaping the evolution of Jakobshavn. We dedicated parts of our paper (both in the main text and the Supplementary Info) to a discussion of those other potential influences.

- Finally, the authors acknowledge on Lines 284-285 that they "... cannot entirely rule out melt serving in some way as a "catalyst" (e.g., by undercutting the front) to accelerate calving, ..." Other statements in the manuscript, however, read as if the role of submarine melting, as presented in our study, has been entirely and conclusively ruled out. Such statements appear on Lines 20-21, 274-275 and 330-331. In light of our responses above, we ask the authors to consider either a) providing evidence that justifies those statements, b) adding nuance to those statements to reflect the fact that the conclusions of our study have not been refuted here, or c) simply removing the parts of those statements that concern our study.

With thanks and best wishes to all,

Ala Khazendar, Josh Willis and Ian Fenty

Jet Propulsion Laboratory, California Institute of Technology

This work was carried out at the Jet Propulsion Laboratory, California Institute of Technology, under a contract with the National Aeronautics and Space Administration. © 2019. California Institute of Technology. Government sponsorship acknowledged.

References:

Bartholomaus, T. C., C. Larsen, and S. O'Neel (2013), Does calving matter? Evidence for significant submarine melt, Earth Planet. Sci. Lett., 380, 21–30.

Benn, D. I., Åström, J., Zwinger, T., Todd, J., Nick, F. M., Cook, S., et al. (2017), Melt‐under‐cutting and buoyancy‐driven calving from tidewater glaciers: New insights from discrete element and continuum model simulations, Journal of Glaciology, 63(240), 691–702.

[Figure]

Carroll, D., et al. (2016), The impact of glacier geometry on meltwater plume structure and submarine melt in Greenland fjords, Geophys. Res. Lett., 43, 9739–9748, doi:10.1002/2016GL070170.

Fried, M. J., G. A. Catania, T. C. Bartholomaus, D. Duncan, M. Davis, L. A. Stearns, J. Nash, E. Shroyer, and D. Sutherland (2015), Distributed subglacial discharge drives significant submarine melt at a Greenland tidewater glacier, Geophys. Res. Lett., 42, 9328–9336, doi: 10.1002/2015GL065806.

Khazendar, A., Fenty, I. G., Carroll, D., Gardner, A., Lee, C. M., Fukumori, I., Wang, O., Zhang, H., Seroussi, H., Moller, D., Noël, B. P. Y., van den Broeke, M. R., Dinardo, S., and Willis, J. (2019), Interruption of two decades of Jakobshavn Isbrae acceleration and thinning as regional ocean cools, Nature Geoscience, 12, 277–283, DOI: 10.1038/s41561-019-0329-3.

Luckman, A., Benn, D. I., Cottier, F., Bevan, S., Nilsen, F., and Inall, M. (2015), Calving rates at tidewater glaciers vary strongly with ocean temperature, Nature Communications, 6, 8566. https://doi.org/10.1038/ncomms9566

O'Leary, M., and Christoffersen, P. (2013), Calving on tidewater glaciers amplified by submarine frontal melting, The Cryosphere, 7(1), 119–128. https://doi.org/10.5194/tc‐7‐119‐2013

Rignot, E., Fenty, I., Xu, Y., Cai, C., and Kemp, C. (2015), Undercutting of marine-terminating glaciers in West Greenland, Geophysical Research Letters, 42(14). https://doi.org/10.1002/2015GL064236.

Todd, J., P Christoffersen, T., Zwinger, P., Råback, N., Chauché, D. Benn, et al. (2018), A full‐Stokes 3‐D calving model applied to a large Greenlandic Glacier, Journal of Geophysical Research: Earth Surface, 123, 410–432. https://doi.org/10.1002/2017JF004349

Please also note the supplement to this comment:
https://www.the-cryosphere-discuss.net/tc-2019-197/tc-2019-197-SC1-
supplement.pdf

---

## Referee Comment (RC1) · Adrian Luckman (Referee) · 27 Sep 2019

This paper presents a comprehensive set of data describing the behaviour of Jakobshavn Isbrae over the last decade. Surface velocities, DEMs, terminus position and ocean temperatures are examined together to investigate recent variability, the role of water temperature and ice-melange on calving, and the potential for the 'ice-cliff instability' to be operating in this location.

I find this paper to be very well presented and written, to make good use of the novel, high quality and comprehensive datasets presented, and to provide a valuable contribution to the literature around calving and outlet glacier stability. The figures are

especially well conceived. I recommend that it be published subject to some minor corrections below.

1) The volume of papers being published in this scientific area has grown very quickly in recent years, so the authors should be forgiven for overlooking some highly relevant works or for missing important citations . Nevertheless, because it directly addresses the issue of melange rigidity on calving, is generally in agreement on the issue, and is also published recently in The Cryosphere, I feel that the recent article by Bevan, myself and others (https://www.the-cryosphere.net/13/2303/2019/), should be mentioned and cited through the text. The authors may also like to consider looking at https://doi.org/10.1016/j.epsl.2015.01.031 which is highly relevant to parts of the discussion on seasonal thinning/thickening.

2) I find the phrase "correct velocity is reported at the wrong location" (page 3 paragraph 1, used twice) to be rather confusing. The issue is important, valid and usually insurmountable, but the way it is described could be clearer. I suggest something like "...so that the .. true geographic location for the retrieved velocity can be displaced by up to 50m from the selected image location leading to a bias in velocity which depends on the velocity gradient" (I'm sure you can do better).

3) Line 109: adding constant → adding a constant

4) Line 155: "there appear to be few, if any, instances of missed detections". This seems unnecessarily vague. Either rigid melange was detected (using the proposed method) or it wasn't - "appear" and "few, if any" make this whole process sound too hit-and-miss (which I don't believe it is).

5) Line 163: "Melange was particularly sparse". I think this needs clarifying since up to now the discussion has been about absence/presence and rigid/non-rigid. What do you mean by sparse (time/space)?. Does the Jakobshavn fjord ever really have open water in it?

[Figure]

6) Line 164: "melange-free". As above. I don't think you mean free of melange, but you probably mean free of rigid melange. I suggest that you make the language a bit tighter here, because it is important.

7) Line 194: meter → meters

8) Line 228: Rather than referring to a "closed white contour" (of which there are several in different panels), I recommend labelling exactly the features you are discussing.

9) Line 415: "more than 130m". This is the first mention of critical cliff heights. I suggest that you refer to a figure here to show that such high cliffs are clearly present in your data.

10) Line 470: "correlate well with . . . AMO". This seemed to be the first mention of such a comparison, so should be in the results or discussion, not left until the conclusion.

Otherwise, great job!

---

## Referee Comment (RC2) · Anonymous Referee #2 · 4 Oct 2019

Joughin et al. presents here a manuscript about the variability of Jakobshavn Isbrae using dense time series of speed and surface elevation over the period 2009-2019. The main conclusion is that the front advances and retreats of Jakobshavn, which is the major forcing for seasonal fluctuations of the glacier (in terms of thickening/thinning and acceleration) is controlled by the rigidity of the mélange and not submarine melt as proposed in previous studies. If true, this could have major implications about the main mechanism leading to rapid retreats of tidewater glaciers around glacier and therefore projection of the evolution of the Greenland Ice Sheets. A other major conclusion concerns the steep ice cliff of Jakobshavn that are not collapsing contrary to the instability mechanism proposed for the rapid retreat of the ice sheet. Finally they provide inter-

esting insights into calving mechanism through the formation of basal crevasses that seems to initiate necking process leading the future large calving.

They are no questions about the quality of data collected and processed here. It is quite a impressive work. The conclusions about the basal crevassing and the absence of ice cliff failure seem robust. My main concern comes from the conclusion on the mélange rigidity versus submarine melt as forcings for the calving rate and so terminus position. Indeed, submarine melt rate stated in Khazendar et al. 2019 are 2 orders of magnitude smaller than those used here. I believe that 8-10 m/yr (line 276) should actually read 8-10 m/day as in Fig. 3a from Khazendar et al.. Values for submarine melt of tidewater glaciers published in other studies (Sciascia et al. 2013, Slater et al. 2018, Sutherland et al. 2019, etc...) are similar to those of Khazendar et al., which consequently, although not perfect, seem realistic. The proper melt rate of 8-10 m/day is therefore about one third of the ice motion at terminus (30-45 m/day) and it becomes obvious that the ice is not replenished "far faster" that the melting (as stated in line 284) and therefore could potentially lead to the undercutting process proposed and sometimes observed in other studies (Sutherland et al. 2019). Considering this, the authors can absolutely not rule out that submarine is a main driver in controlling the terminus position. In addition, the observations of "strong" mélange during period of advance and weak mélange during retreat could be just coincidental as ice mélange is most probably weak during period of high submarine melt and vice-versa. The discussion would also be strengthened if recent modeling studies and mechanisms that would prevent ice calving in the presence of ice mélange were included (such as Krug et al. 2015). The last comment is about the implications for other glaciers in Greenland that is not mentioned in the paper. The presence of such "thick" mélange is particular to Jakobshavn Isbrae, where icebergs are well confined in a long fjord. Glaciers along NW coast also display seasonal variations but it is less obvious that such a thick ice mélange is present for these glaciers that are more open on the ocean. Would the presence of relatively thin sea ice also have the same impact on the calving rate ?

That said I still believe that the discussion about the mélange rigidity is interesting and it is possible that both mechanisms (submarine melt and ice mélange) are influencing the calving rate. I appreciate the effort made for gathering and processing all these datasets and the interesting conclusions on the formation of basal crevasse and ice cliff failure. I would therefore recommend revision of the paper according the above comments and much milder conclusion on the influence of submarine melt vs ice mélange concentration.

Khazendar, A., Fenty, I. G., Carroll, D., Gardner, A., Lee, C. M., Fukumori, I., et al. (2019). Interruption of two decades of Jakobshavn Isbrae acceleration and thinning as regional ocean cools. Nature Geoscience, 12(4), 277-283. https://doi.org/10.1038/s41561-019-0329-3

Krug, J., Durand, G., Gagliardini, O., and Weiss, J.: Modelling the impact of submarine frontal melting and ice mélange on glacier dynamics, The Cryosphere, 9, 989-1003, https://doi.org/10.5194/tc-9-989-2015, 2015.

Sciascia, R., Straneo, F., Cenedese, C., and Heimbach, P. ( 2013), Seasonal variability of submarine melt rate and circulation in an East Greenland fjord, J. Geophys. Res. Oceans, 118, 2492- 2506, doi:10.1002/jgrc.20142.

Slater, D. A., Straneo, F., Das, S. B., Richards, C. G., Wagner, T. J. W., & Nienow, P. W. (2018). Localized Plumes Drive Front‐Wide Ocean Melting of A Greenlandic Tidewater Glacier. Geophysical Research Letters, 45(22), 12,350-12,358. https://doi.org/10.1029/2018GL080763

Slater, D. A., Straneo, F., Das, S. B., Richards, C. G., Wagner, T. J. W., & Nienow, P. W. (2018). Localized plumes drive front-wide ocean melting of a Greenlandic tidewater glacier. Geophysical Research Letters, 45, 12,350-12,358. https://doi.org/10.1029/2018GL080763

---

## Author Comment (AC1) · 8 Nov 2019

**Response to Comment 1**

*Thanks for these comments, which greatly improved the paper.*

This Short Comment focuses exclusively on correcting some inaccurate representations of our findings that we reported in Khazendar et al. (2019). We gratefully thank the authors in advance for their kind consideration.

- The authors incorrectly characterize one of the main conclusions of our study in Lines 276-282: "Maximum melt rates for 2012 to 2015 are estimated to be    8-10 m/yr for concentrated plumes with limited spatial extent, or about a factor 3 less if the melt water emerges with a uniform distribution from beneath the grounded ice (Khazendar et al., 2019). Whether concentrated or evenly distributed, the factor-of-3 reduction should roughly represent the average melt rate across the terminus. Thus, scaling the plume rates from Khazendar et al. (2019) by a factor of 3 yields approximate average melt between    3.5 m/yr during the summers with warmest water and    1.9 m/yr during the summers when the water was coolest." In their argument, the authors take our values of maximum melt rates and translate them to lower mean rates.

*Aside from our typo (m/yr in place of m/d – force of habit), we stand by our statement, which is fully consistent with the results stated in Khazendar et al., 2019. We did work, however, to make the language a bit more precise, as in fact we overestimated the average melt for the plume case in the earlier draft. If we assume that the plume is 100-150 m wide, and average the 10.5 m/day maximum rate across the ~4-km –wide terminus (not in depth), then the average  is < 1 m/day (we didn't compute the exact rate because it depends on the exact width of the plume and the terminus width varies – but these example values yield a width average of ~0.4/day, so saying < 1 m/day is appropriate).*

*We also now make clear we are referring to width-averaged, not depth-averaged rates. It is important to note that our interpretation of what is said in Khazendar et al.  is that the maximum melt rates apply to a single narrow plume (From Khazendar et al "A point-source subglacial plume at the front of Jakobshavn is modelled using ocean temperature data collected in 2019"). So, one gets the most bang for their subglacial melt volume with a uniform distribution, which yields a width-averaged maximum rate of 3.5 m/d after apply the scaling factor provided by Khazendar et al ("During summer, if the subglacial discharge is evenly distributed across the width of the terminus as a line plume, instead of emerging from a single subglacial conduit, melting rates are reduced by roughly a factor of 3"). To the best of our knowledge, these numbers follow directly from the numbers and assumptions in Khazendar et al., and this is consistent with other plume studies in the literature.  For example, experimenting with the simple model of Xu et al, 2013, it's clear that the*

*maximum total melt is achieved when the melt emerges as a uniform line plume. Taking all of this into account, 3.5 m/day is the maximum melt rate in width-averaged sense based on the Khazendar et al results , which, as we point out, is small relative to a 45 m/day advance. Even a concentrated melt rate of 10.5 m/d is small relative to the advance rate. Moreover, this rate applies to a narrow plume that would be partly offset by the bridging effects of the nearby ice (e.g., for 100-m wide tunnel in a 4-km wide ice front). The now cited Todd et al papers strengthen our assertion.*

**Yet, we opted to present the maximum melting rates purposefully.**

> *We realize that, but we disagree with that decision, and our statements provide relevant context in which to interpret those numbers. It is disappointing that the melt rates presented in Khazendar et al are not more clearly identified as melt rates for one narrow (100-m scale) location in an ~4000 m ice front (one needs to read the Methods and some of the cited papers to fully appreciate that).*

**For deep glaciers such as Jakobshavn Isbrae, ocean-induced melting at the front tends to reach its maximum value within 100 m of the grounding line up the face of the glacier (Carroll et al., 2016).**

> *Yes, but the single plume modeled in Khazendar et al. is most likely to produce a narrow cleft in a wide terminus, as noted above. To make the point clear, we now state the following:*

"Maximum melt rates for 2012 to 2015 are estimated to be ~8-10.5 m/d for a concentrated plume with limited spatial extent (~100-150 m) at the terminus of Jakobshavn Isbrae, which when averaged across the width of the terminus face gives a mean rate of <1 m/d (Khazendar et al., 2019). Due to the non-linear relation between melt and subglacial melt discharge (Xu et al., 2013), maximum aggregate melt should be achieved when the subglacial melt emerges uniformly from beneath the terminus. In this case, melt rates are about a factor 3 less than the corresponding plume rates (Khazendar et al., 2019), yielding a maximum width-averaged rate ~3.5 m/d during the recent warm period. Similarly, the maximum width-averaged melt is ~1.9 m/d during cool periods, based on an ~5.7 m/d maximum plume rate. Note all rates reflect the maximum rate at some depth, so the depth-averaged rates should be somewhat smaller (Carroll et al., 2016; Khazendar et al., 2019). It is also important to note that much of the oceanic heat in the fjord goes into melting icebergs (Moon et al., 2018), so these values may be biased high. During the summer, the terminus advances at ~30-45 m/d, so that ice is replenished far faster than it is removed via submarine melting (<1–3.5 m/d) (Joughin et al., 2012a). "

> *While there is a different emphasis, we believe this is an accurate reflection of the results presented in in Khazendar et al. We are happy to correct any factual errors in this statement.*

**This enhanced melting has been observed to produce widespread undercutting of the glacier fronts (Fried et al., 2015; Rignot et al., 2015), which could lead to increased calving, frontal retreat and reduced resistance to flow.**

*These papers do demonstrate some undercutting, but do not demonstrate an accelerated rate of calving (Rink and Store are two of the more stable glaciers in Greenland). Moreover, the glacier examined by Fried et al. is moving an order of magnitude slower than Jakobshavn, so the undercutting is of more comparable magnitude to the ice speed.*

**Observations and theoretical work have suggested that calving can be a direct response to undercutting at the front (Bartholomaus et al., 2013; Luckman et al., 2015) and that submarine melting and undercutting can contribute to calving that is several times the melting rate (O'Leary and Christoffersen, 2013; Benn et al., 2017; Todd et al., 2018).**

*The Luckman et al. paper references a glacier where the terminus speed is comparable to the ablation rate. They acknowledge that this process likely dominates at slower glaciers rather than faster glaciers. The Bartholomaus et al. paper is for a slower glacier in much warmer water and a much shallower terminus subject to fairly different calving dynamics. And despite melt rates of 9–17 m/d (average), the glacier is advancing. While a nice piece of work, the O'Leary and Christoffersen paper uses a 2-D model and focuses on the shifting of the stress concentration inland, which does not necessarily increase calving (especially for the near-flotation case). We did add a reference to this paper. The more realistic cases that Krug et al. and Todd et al. model show relatively modest sensitivity to melt, and a far greater sensitivity to mélange. When the mélange is included in the Todd et al. model, plume melt rates, for the most most part, are not significant until they reach very large rates of ~24 m/day. In response to this comment and those by the other reviewers, we added the following to make these points:*

"While we cannot entirely rule out melt serving in some way as a "catalyst" (e.g., by undercutting the front) to influence calving, a shift in average melt rate from 3.5 to 1.9 m/d (e.g., average melt decreased by 1.6 m/d) over a few months of the year should not drastically slow the rate of retreat and speedup for a glacier that moves at 30-45 m/d. For those glaciers where undercutting has been observed to have a substantial effect, the melt rate is comparable to the terminus advance rate (Luckman et al., 2015), unlike the case for Jakobshavn Isbrae where width-averaged melt rates are an order of magnitude slower. While a 2-D model does suggest that even modest undercutting may have some effect (O'Leary and Christoffersen, 2012), the main effect for cases near flotation is to shift a relatively weak, broad stress peak inland. A more complex time-dependent model that includes calving with damage indicates that the effect mélange on seasonal variation in terminus position and speed is far greater than that of melt undercutting (Krug et al., 2015). Neither model accounts for basal crevassing, which can be important for calving near flotation (Van Der Veen, 1998). In a full 3D model that includes both basal and surface crevassing, plume melt rates of 12 m/d in combination with uniformly distributed melt rates 3.1 m/d produce little seasonally enhanced calving (Todd et al., 2018). It is only when plume melt rates are increased to ~24 m/d that is there a substantial effect for a glacier flowing more slowly (12–14 m/d) than Jakobshavn Isbrae (Todd et al., 2019). As with the 2-D model (Krug et al., 2015), the 3-D model produces a pronounced variation in terminus position and speed in response to seasonal mélange forcing, consistent with our observations."

**Therefore, rather than scaling our quoted melt rates down, it is more likely that the melt rates should be scaled up to represent their potential impact on the calving rate.**

> *Please see arguments above that address this issue. In summary, we scale only using the Khazendar et al. supplied factor and average results across the terminus to compare the rates with the terminus advance rates.*
>
> *The commenters' argument suggests that we should interpret the referenced papers as showing that undercutting is the dominant factor in calving of Jakobshavn Glacier, and that we should scale the results in Khazendar et al by an arbitrary factor to demonstrate that this is so. We are reluctant to engage in this kind of circular reasoning, particularly because our reading of those papers has not provided a convincing argument for the commenters' premise.*

**Furthermore, we emphasized in our paper that while we found that the glacier's thickness changes had a strong correlation with ocean temperature variability, the former was even more strongly correlated with the variability in submarine melting rates.**

> *Since we are both arguing for processes modulated by the ocean temperature, we don't dispute the correlation. But unlike the annual spring elevation data in Khazendar et al., our conclusions are based on much more temporally dense elevation data. These data indicate that thickening primarily commences during winter, when the mélange processes come into play.*

**- The authors then continue on Lines 289-291 with another argument to justify rejecting the relevance of submarine melting: "Although submarine melt should have been substantially reduced in the summer of 2016 (Khazendar et al., 2019), the maximum retreat was virtually identical to that of the four prior years and speeds were only slightly reduced, suggesting melt is not directly controlling retreat."**

**This is a mischaracterization of our data and approach.**

> *This statement is meant to characterize our data, not those of Khazendar et al, and does not include any mention of their approach. Nonetheless it is not inconsistent with the data presented by Khazendar et al.*

**It is clear from the Davis Strait mooring data (Figure 3 in Khazendar et al., 2019) that cold water started arriving in Disko Bay in June or July of 2016, and in fact Figure 2 of Joughin et al. shows that glacier speeds in 2016 at Tmax-1km, M6 and M9 all experienced the smallest increases between the spring minimum and the summer peak than any other summer since 2011. Our Figure 3 shows the same.**

> *We state:*

"In contemplating whether submarine melt, particularly in summer, might drive the observed retreat and speedup, it is important to consider the relative timing of the recent changes. As

Figure 6 shows, the colder water first appears in the fjord in the summer of 2016 (Khazendar et al., 2019). While the summer 2016 speeds are moderately slower than prior summers (2012–2015), this decline is not far off a general trend of declining summer peaks following the summer of 2012. During the period when the position of maximum summer retreat was relatively stable (2012–2016), the declining summer peaks were likely a consequence of the evolving geometry that shallowed slopes over time in the near-terminus region (Fig. 3). It is also important to note that to the extent that submarine melt may influence speed, it is through changes in the calving rate that influence terminus position, which then alter speed (e.g., Fig. 5). Since the position of minimum retreat is virtually the same in 2016 as in the prior four summers, the cooler water does not appear to have suppressed calving that summer."

*Our main point was that things really begin change in the winter of 2016/2017, when there is a strong terminus advance.*

**As we state in our paper, the ocean properties and subglacial discharge volumes we use in calculating submarine melting rates are from the summer of each year, while the thickness changes of Jakobshavn are from the following spring, when the altimetry data were acquired. Regarding the year 2016, this is what we stated: "Most prominently, the sharp drop in ocean temperatures in 2016 and 2017 by 2 °C relative to the peak temperature in 2014 corresponds to the slowing and dramatic thickening of the glacier in 2017 and 2018." Indeed, our observations (Fig. 3 in Khazendar et al., 2019) show that the flow speed of Jakobshavn starts a significant slowdown in the summer of 2016, around the time of the observation of the large drop in ocean temperatures and submarine melting rates. The glacier then reaches its slowest flow speeds in the spring of 2017, coinciding with our measurement of significant thickening. The flow speed then stages only a weak recovery in the summer of 2017. This pattern is also shown by the data in Figure 2 of Joughin et al.**

*Our text does not contradict that interpretation, except that we feel for the most part the slowdown commences in winter 2016. As noted above, the summer 2016 was only modestly slower. Moreover, the retreat, which governs speed, was identical to that of the summers with warm water (see revised text above).*

**Our rendering of the events and their relative timing is consistent, so we request that authors remove the text on Lines 289-291 in its current form as it misconstrues our findings.**

*Our statement was : "Although submarine melt should have been substantially reduced in the summer of 2016 (Khazendar et al., 2019), the maximum retreat was virtually identical to that of the four prior years and speeds were only slightly reduced, suggesting melt is not directly controlling retreat."*

*We disagree with the commentors' assertion that this statement misconstrues their findings. We offer the following justification:*

*1) Figure 3 of Khazendar et al shows a reduction in melt for summer 2016*

*2) Our data show the minimum retreat is similar to that in the previous few years. The only point in question is whether the slowdown actually began in summer 2016, and we have a different interpretation – we have added some text to make our point more clear (see above).*

*As an aside, we note that during the 5-year period with similar terminus extent, the second slowest observed summer speeds occurred during 2014 (about 0.25 m/d slower than 2016 in Fig 3 from Khazendar et al). Yet this is a summer of maximum melt rates (approximately equivalent to 2012 melt rates – the year with fastest observed summer speeds). So, in terms of speed, the correlation with melt rates is weak.*

*We feel our statement and interpretation are valid, and acknowledge that there is room for continued debate and a need for continued observation and analysis.*

More generally, we aimed to be careful in framing the conclusions of our study as not to claim that ocean temperature variability and submarine melting are the sole explanations of Jakobshavn's dynamic evolution. We wrote that we "find the evidence sufficient to conclude that ocean temperature variability, through its influence on submarine melting rates, has been a main, and sometimes dominant, factor in shaping Jakobshavn Isbrae's interannual dynamic evolution since the disintegration of the ice shelf in 2003." We feel this conclusion holds without us having to dismiss the possibility that other processes might also have had a role in shaping the evolution of Jakobshavn. We dedicated parts of our paper (both in the main text and the Supplementary Info) to a discussion of those other potential influences.

*And in our paper, we argue that the mélange is likely the "main, and sometimes dominant" forcing that controlled Jakobshavn's behavior over the last decade. And we too did not rule out other processes, such as melt.*

- Finally, the authors acknowledge on Lines 284-285 that they ". . . cannot entirely rule out melt serving in some way as a "catalyst" (e.g., by undercutting the front) to accelerate calving, . . ." Other statements in the manuscript, however, read as if the role of submarine melting, as presented in our study, has been entirely and conclusively ruled out.

*We feel that our approach is more productive in advancing the scientific debate than was the treatment of mélange in Khazendar et al.  There, the only reference to mélange is: "The roles of ice mélange on interannual timescales[4,34–36], and that of cryo-hydrologic warming[37,38], have yet to be elucidated." In fact, a number of papers had, at the time, explored the importance of mélange in controlling calving and glacier advance (e.g. Todd et. al, 2018, and Krug et al, 2015). While the commenters may not agree with our opinion, we feel it is important to continue the open discussion on the relative importance of the different mechanisms.*

**Such statements appear on Lines 20-21, 274-275 and 330-331. In light of our responses above, we ask the authors to consider either a) providing evidence that justifies those statements, b) adding nuance to those statements to reflect the fact that the conclusions of our study have not been refuted here, or c) simply removing the parts of those statements that concern our study.**

> *A) We improved our arguments as described above.*
>
> *B) We are not "refuting" the conclusions of Khazendar et al. We presented a different hypothesis. This process is fundamental to how science works, and ongoing observation and analysis will reveal which hypothesis (or some combination) is correct. Our paper is very carefully worded to make clear we are hypothesizing with justification based on observations and the literature (26 instances of "may", 9 instances of "appears", 8 instances of "suggest", 16 instances of "likely"), indicating due diligence with respect to ensuring our points are nuanced.*
>
> *C) We reworded several statements as described above.*

With thanks and best wishes to all,
Ala Khazendar, Josh Willis and Ian Fenty
Jet Propulsion Laboratory, California Institute of Technology

This work was carried out at the Jet Propulsion Laboratory, California Institute of Technology, under a contract with the National Aeronautics and Space Administration. © 2019. California Institute of Technology. Government sponsorship acknowledged.

References:

Bartholomaus, T. C., C. Larsen, and S. O'Neel (2013), Does calving matter? Evidence for significant submarine melt, Earth Planet. Sci. Lett., 380, 21–30.

Benn, D. I., Åström, J., Zwinger, T., Todd, J., Nick, F. M., Cook, S., et al. (2017), Melt‐under‐cutting and buoyancy‐driven calving from tidewater glaciers: New insights from discrete element and continuum model simulations, Journal of Glaciology, 63(240), 691–702.

Carroll, D., et al. (2016), The impact of glacier geometry on meltwater plume struc- ture and submarine melt in Greenland fjords, Geophys. Res. Lett., 43, 9739–9748, doi:10.1002/2016GL070170.

Fried, M. J., G. A. Catania, T. C. Bartholomaus, D. Duncan, M. Davis, L. A. Stearns, J. Nash, E. Shroyer, and D. Sutherland (2015), Distributed subglacial discharge drives significant submarine melt at a Greenland tidewater glacier, Geophys. Res. Lett., 42, 9328–9336, doi: 10.1002/2015GL065806.

Khazendar, A., Fenty, I. G., Carroll, D., Gardner, A., Lee, C. M., Fukumori, I., Wang, O., Zhang, H., Seroussi, H., Moller, D., Noël, B. P. Y., van den Broeke, M. R., Dinardo, S., and Willis, J. (2019), Interruption of two decades of Jakobshavn Isbrae acceler- ation and thinning as regional ocean cools, Nature Geoscience, 12, 277–283, DOI: 10.1038/s41561-019-0329-3.

Luckman, A., Benn, D. I., Cottier, F., Bevan, S., Nilsen, F., and Inall, M. (2015), Calving rates at tidewater glaciers vary strongly with ocean temperature, Nature Communica- tions, 6, 8566. https://doi.org/10.1038/ncomms9566

O'Leary, M., and Christoffersen, P. (2013), Calving on tidewater glaciers amplified by submarine frontal melting, The Cryosphere, 7(1), 119–128. https://doi.org/10.5194/tcăŘ̌7ăŘ̌119ăŘ̌2013

Rignot, E., Fenty, I., Xu, Y., Cai, C., and Kemp, C. (2015), Undercutting of marine- terminating glaciers in West Greenland, Geophysical Research Letters, 42(14). https://doi.org/10.1002/2015GL064236.

Todd, J., P Christoffersen, T., Zwinger, P., Råback, N., Chauché, D. Benn, et al. (2018), A fullăŘ̌Stokes 3ăŘ̌D calving model applied to a large Green- landic Glacier, Journal of Geophysical Research: Earth Surface, 123, 410–432. https://doi.org/10.1002/2017JF004349

---

## Author Comment (AC2) · 12 Nov 2019

**Response to Reviewer 1 (Adrian Luckman)**

*Thanks for these comments, which greatly improved the paper.*

**This paper presents a comprehensive set of data describing the behaviour of Jakobshavn Isbrae over the last decade. Surface velocities, DEMs, terminus position and ocean temperatures are examined together to investigate recent variability, the role of water temperature and ice-melange on calving, and the potential for the 'ice-cliff instability' to be operating in this location.**

**I find this paper to be very well presented and written, to make good use of the novel, high quality and comprehensive datasets presented, and to provide a valuable contribution to the literature around calving and outlet glacier stability. The figures are especially well conceived. I recommend that it be published subject to some minor corrections below.**

*Thanks for the commentary – no specific action so no change.*

**The volume of papers being published in this scientific area has grown very quickly in recent years, so the authors should be forgiven for overlooking some highly relevant works or for missing important citations . Nevertheless, because it directly addresses the issue of melange rigidity on calving, is generally in agreement on the is- sue, and is also published recently in The Cryosphere, I feel that the recent article by Bevan, myself and others (https://www.thecryosphere.net/13/2303/2019/), should be mentioned and cited through the text. The authors may also like to consider looking at https://doi.org/10.1016/j.epsl.2015.01.031 which is highly relevant to parts of the discussion on seasonal thinning/thickening.**

*We did not include the Helheim paper along with a good number of other papers that could be considered relevant (I think we are at 59 references). The Kanger paper, however, was highly relevant and we thank you for calling it to our attention. We have also added another half dozen or so references in response to other comments.*

**I find the phrase "correct velocity is reported at the wrong location" (page 3 paragraph 1, used twice) to be rather confusing. The issue is important, valid and usually insurmountable, but the way it is described could be clearer. I suggest something like "...so that the .. true geographic location for the retrieved velocity can be displaced by up to 50m from the selected image location leading to a bias in velocity which depends on the velocity gradient" (I'm sure you can do better).**

*Not sure if this does the trick, but we have reworded to "With the relatively accurate DEMs we used, the errors should tend toward the low end of this range in most instances. Exceptions may occur where large temporal fluctuations in slope occur near the terminus as*

discussed below. Another issue is geolocation error since errors in DEM elevation directly translate into horizontal position errors. Although we generally update the DEM annually, large intra-annual changes can introduce absolute geolocation errors of up to ~50 m. In such cases, an otherwise correct velocity measurement is posted at the wrong location, which in a gridded product is equivalent to an additional source of velocity error, especially where velocity gradients are strong. This problem can be exacerbated when comparing data acquired from differing imaging geometries (e.g., from ascending and descending passes), since the DEM-induced location shifts can occur in opposing directions to produce a relative geolocation error of ~100 m.*"*

**Line 109: adding constant → adding a constant**

*Done.*

**Line 155: "there appear to be few, if any, instances of missed detections". This seems unnecessarily vague. Either rigid melange was detected (using the proposed method) or it wasn't - "appear" and "few, if any" make this whole process sound too hit-and-miss (which I don't believe it is).**

*Actually "missed detections" was used purposely. There are two types of errors "missed detections" or "false alarms" (aka false positives). We are not particularly worried about the latter – its hard to get a coherent match when the data are incoherent. We do worry about the former, which is why we did some visual inspection. We did remove "appear" and changed to "are" to make a little less vague.*

**5) Line 163: "Melange was particularly sparse". I think this needs clarifying since up to now the discussion has been about absence/presence and rigid/non-rigid. What do you mean by sparse (time/space)?. Does the Jakobshavn fjord ever really have open water in it?**

*Good point, especially re sparse. Changed to "*The occurrence of rigid mélange was particularly infrequent in both 2011 and 2012*"*

**6) Line 164: "melange-free". As above. I don't think you mean free of melange, but you probably mean free of rigid melange. I suggest that you make the language a bit tighter here, because it is important.**

*Agreed. Changed to "*rigid-melange-free*"*

**7) Line 194: meter → meters**

*Done*

**8) Line 228: Rather than referring to a "closed white contour" (of which there are several in different panels), I recommend labelling exactly the features you are discussing.**

*Added a blue diamond in one panel to serve as reference point. Updated text accordingly.*

**9) Line 415: "more than 130m". This is the first mention of critical cliff heights. I suggest that you refer to a figure here to show that such high cliffs are clearly present in your data.**

*Added "*(see Fig. 7)*", which most clearly indicates this is the case.*

**10) Line 470: "correlate well with . . . AMO". This seemed to be the first mention of such a comparison, so should be in the results or discussion, not left until the conclusion.**

*We are inclined to leave this here. In the discussion we more focus on the relation between water temperature and stability without getting into the climate aspect. In the conclusion we are more presenting a broader outlook for going forward and this seems a good note on which to end the paper.*

**Otherwise, great job!**

*Thanks*

---

## Author Comment (AC3) · 12 Nov 2019

**Response to Reviewer 2**

*Thanks for these comments, which greatly improved the paper.*

Joughin et al. presents here a manuscript about the variability of Jakobshavn Isbrae using dense time series of speed and surface elevation over the period 2009-2019. The main conclusion is that the front advances and retreats of Jakobshavn, which is the major forcing for seasonal fluctuations of the glacier (in terms of thickening/thinning and acceleration) is controlled by the rigidity of the mélange and not submarine melt as proposed in previous studies. If true, this could have major implications about the main mechanism leading to rapid retreats of tidewater glaciers around glacier and therefore projection of the evolution of the Greenland Ice Sheets. Another major conclusion concerns the steep ice cliff of Jakobshavn that are not collapsing contrary to the instability mechanism proposed for the rapid retreat of the ice sheet. Finally, they provide interesting insights into calving mechanism through the formation of basal crevasses that seems to initiate necking process leading the future large calving.

*Summary comment, no action taken.*

They are no questions about the quality of data collected and processed here. It is quite a impressive work. The conclusions about the basal crevassing and the absence of ice cliff failure seem robust. My main concern comes from the conclusion on the mélange rigidity versus submarine melt as forcings for the calving rate and so terminus position. Indeed, submarine melt rate stated in Khazendar et al. 2019 are 2 orders of magnitude smaller than those used here. I believe that 8-10 m/yr (line 276) should actually read 8-10 m/day as in Fig. 3a from Khazendar et al..

*Yes – this was a typo by an author long used to working in units of m/yr. These were corrected to m/day throughout the text.*

Values for submarine melt of tidewater glaciers published in other studies (Sciascia et al. 2013, Slater et al. 2018, Sutherland et al. 2019, etc. . .) are similar to those of Khazendar et al., which consequently, although not perfect, seem realistic.

*Our argument is not with the melt rate values, but rather how they are presented. We offer additional context and evaluate the Khazendar et al. numbers. For example, while their results show that while a single plume may have localized melt rates of ~10 m/d, this would produce a ~100-200 m wide "slot" in a 4-km-wide by ~1 km thick ice front. The maximum localized plume melt rate of ~10 m/day averaged across the full width is less than 1 m/d ((~100 m wide plume * 10 m/day)/4000 m is ~0.25 m/day).*

*Due to the non-linearity between subglacial melt and terminus melt, you will get more terminus melt by spreading the same subglacial meltwater volume across*

*multiple plumes, tending toward a maximum for a uniform distribution.  For example, on another large glacier with similar ocean temperatures, Moon et al. (2018) estimate melt rates of 38 m^3/s for 10 plumes at Helheim Glacier.  If we assume that all of the melt is from subglacial discharge at Helheim (ignoring input to this number from the two smaller adjacent glaciers), a  back-of-the-envelope estimate for melt rates is ~1.1 m/day for a 6000 m wide terminus with most of the melting on the lower 500 m (38/(6000\*500) \* 86400 m^3/s m^-2 \* s/d), which is small relative to the advance rate of either Helheim or Jakobshavn.*

*In any event, whether single-plume (least total melt) or evenly distributed subglacial melt (maximum total melt) the maximum estimate for width-averaged melt for Jakobshavn is still <3.5 m/d based on the Khazendar et al provided scale factor, which is small relative to the observed terminus speed of 30–45 m/d.*

*So, we did not change the text in direct response to this comment, but we improved the melt discussion in response to other comments.*

**The proper melt rate of 8-10 m/day is therefore about one third of the ice motion at terminus (30-45 m/day) and it becomes obvious that the ice is not replenished "far faster" that the melting (as stated in line 284) and therefore could potentially lead to the undercutting process proposed and sometimes observed in other studies (Sutherland et al. 2019).**

*As noted above, the 8-10 m/day melt rate estimate applies to a ~100 m wide plume, which could produce a cavity of about that width (or perhaps a factor of ~2x wider), but this would result in one "slot" carved in 4-km wide ice cliff.*

*Todd et al. (2018) show that at slightly larger plume rates COMBINED with 3.1 meters of uniformly distributed melt there is very little seasonal effect on the speed and terminus position of a slower glacier. They also show that the influence of the melt plume/evenly distributed melt diminishes once the stabilizing influence of mélange is included in the model – compare case 011 (plume 12 m/d and 3.1 m/d evenly distributed melt) with 111 (melt as in 011 but with mélange) in their Fig. 6. There is no evidence to support melt rates any larger than those used by Todd et al. (2018) at this time and they may be overestimates (see also next paragraph).*

*The paper by Sutherland et al finds both some undercutting and some overcutting, but doesn't comment on finding related calving enhancement. A central point they make is that distributed (ambient) melting is greater than plume theory would predict, which is relevant to our results.  There are, however, some important differences.  For the LeConte glacier the water at depth ranges from ~4 to 7.5 deg, compared with ~3.5degC for Jakobshavn. For*

*the 4-degC case, which still exceeds the water temperature at Jakobshavn, the average melt rate is 0.9 m/d (compare with back of the envelope calculation above), which doesn't change our central point that rates of this magnitude are small relative to the calving rate. Another central point of the Sutherland et al paper is that flux gate methods tend to overestimate melt by about a factor of 2, which would mean earlier melt-rate flux-gate estimates for Greenland may be too high by about a factor of two (e.g., Rignot et al,2010; Xu et al, 2013. For the latter paper, if such a bias exists and could be removed it would bring their model results into better agreement with their observations). The upshot though is that Todd et al., 2018 already include both plume and distributed melt at rates that probably exceed that for Jakobshavn based on our interpretation of Sutherland et al (e.g., 4deg -> 0.9 m/d) and other results. Hence, Sutherland et al further support our hypothesis rather than weaken it. We do not cite this paper because a comprehensive review of the various melt rates methods is beyond the scope of our paper. (We do cite work that includes Sutherland – Moon et al 2018 – that looks at Greelandic glacier – see Helheim discussion above).*

Considering this, the authors can absolutely not rule out that submarine is a main driver in controlling the terminus position. In addition, the observations of "strong" mélange during period of advance and weak mélange during retreat could be just coincidental as ice mélange is most probably weak during period of high submarine melt and vice-versa. The discussion would also be strengthened if recent modeling studies and mechanisms that would prevent ice calving in the presence of ice mélange were included (such as Krug et al. 2015).

*We do not attempt to offer absolute claims, but present evidence that supports the hypothesis that mélange has greater influence than submarine melting for Jakobshavn during the period of our study.*

*There is some degree of correspondence employed in both arguments. We feel that the timing of slowdown and the onset of melange rigidity makes a more compelling case. As does similar degree of retreat the first summer the water was cold. And we believe that the model studies we cite better support our hypothesis. As do similar correspondences on other glaciers (e.g., Bevan et al). Neither hypothesis can be fully proved at this point – but we feel ours better fits the observations and other results in the current literature.*

*Thanks for bringing our attention to this paper, which supports our hypothesis. We added "*A more complex time-dependent model that includes calving with damage indicates that the effect mélange on seasonal variation in terminus position and speed is far greater than that of melt undercutting (Krug et al., 2015).*" We also add text citing other modeling studies.*

The last comment is about the implications for other glaciers in Greenland that is not mentioned in the paper. The presence of such "thick" mélange is particular to

**Jakobshavn Isbrae, where icebergs are well confined in a long fjord. Glaciers along NW coast also display seasonal variations but it is less obvious that such a thick ice mélange is present for these glaciers that are more open on the ocean. Would the presence of relatively thin sea ice also have the same impact on the calving rate ?**

> *This a good point and a good question. Certainly, work by Reeh suggests that for some glaciers (e.g., NE Greenland), sea ice alone can suppress calving. But that work applied to rather thin, extended ice shelves. There certainly is winter mélange near many glaciers along the NW coast (e.g., see Moon et al. papers on subject), but the record of its rigidity is less certain than in regions that were imaged more frequently (like Jakobshavn).*

> *A detailed analysis of this question is beyond the scope of this paper. To acknowledge this point, however, we added as the paper's last sentence "*While our results should be applicable to glaciers with high calving rates that yield a thick mélange [Bevan et al., 2019; Kehrl et al., 2017], more work is needed to understand the influence of thinner mélange on smaller glaciers that calve less rapidly*."*

**That said I still believe that the discussion about the mélange rigidity is interesting and it is possible that both mechanisms (submarine melt and ice mélange) are influencing the calving rate. I appreciate the effort made for gathering and processing all these datasets and the interesting conclusions on the formation of basal crevasse and ice cliff failure. I would therefore recommend revision of the paper according the above comments and much milder conclusion on the influence of submarine melt vs ice mélange concentration.**

> *We agree this not the final word on the subject, but we disagree with the reviewer's recommendation. We present a hypothesis that can be tested in the future. Our more extensive literature review, completed in response to reviewer comments, has strengthened support for this hypothesis.*

> *We feel that the text and qualifying language (e.g., "may", "suggests") in the paper emphasize that this is a testable hypothesis supported by a large body of observational and modelling work. For example, our statement in the abstract says "*Thus, along with the relative timing of the seasonal slowdown, our results **_suggest_** that the ocean's dominant influence on Jakobshavn Isbrae is through its effect on winter mélange rigidity, rather than summer submarine melting*.", which we note is comparable to the level of certainty in the Khazendar et al abstract. As we note in the response to the Khazendar et al comment, additional observations and study will provide further evaluation of these competing hypotheses.*

Khazendar, A., Fenty, I. G., Carroll, D., Gardner, A., Lee, C. M., Fukumori, I., et al. (2019). Interruption of two decades of Jakobshavn Isbrae accelera- tion and thinning as regional ocean cools. Nature Geoscience, 12(4), 277-283. https://doi.org/10.1038/s41561-019-0329-3

Krug, J., Durand, G., Gagliardini, O., and Weiss, J.: Modelling the impact of submarine frontal melting and ice mélange on glacier dynamics, The Cryosphere, 9, 989-1003, https://doi.org/10.5194/tc-9-989-2015, 2015.

Sciascia, R., Straneo, F., Cenedese, C., and Heimbach, P. ( 2013), Seasonal variability of submarine melt rate and circulation in an East Greenland fjord, J. Geophys. Res. Oceans, 118, 2492- 2506, doi:10.1002/jgrc.20142.

Slater, D. A., Straneo, F., Das, S. B., Richards, C. G., Wagner, T. J. W., & Nienow, P. W. (2018). Localized Plumes Drive Front ˇWide Ocean Melting of A Green- landic Tidewater Glacier. Geophysical Research Letters, 45(22), 12,350-12,358. https://doi.org/10.1029/2018GL080763

Slater, D. A., Straneo, F., Das, S. B., Richards, C. G., Wagner, T. J. W., & Nienow, P. W. (2018). Localized plumes drive front-wide ocean melting of a Greenlandic tidewater glacier. Geophysical Research Letters, 45, 12,350-12,358. https://doi.org/10.1029/2018GL080763

---

## Author Response (AR1)

This is the marked-up version of all the changes, which are described point by point in the posted online responses to the reviewers and commenters. After we uploaded comments, we did a final edit, so there may be minor changes between what is presented below and what is quoted in the comments. The main substance of the changes should be common to those comments and this marked up version.

[revised manuscript text omitted]